# GACET: A Graph-Aware Cross-Domain EEG Transformer with the CD-MWL Benchmark

## Abstract

Robust generalization of brain state decoding across days remains a grand challenge for brain-computer interfaces (BCIs), throttling the real-world deployment of applications like mental-workload (MWL) estimation. This long-standing problem has been difficult to address, largely due to the scarcity of public corpora suitable for rigorous long-term evaluation. To establish the first robust benchmark for this challenge, we introduce CD-MWL, the longest public MWL dataset to date: 42 hours of EEG from 14 participants over three days. Building on this benchmark, we argue that true generalization requires models that are not only high-performing but also neuroscientifically plausible. We therefore propose GACET, a Graph-Aware Cross-domain EEG Transformer that achieves superior generalization by dynamically learning the brain's underlying functional connectivity in a neuroscientifically plausible manner. Through topology-aware message passing on this learned graph, GACET not only delivers significant performance gains over all state-of-the-art methods but also provides a transparent window into its decision-making process. Crucially, we demonstrate that the model's learned connectivity patterns align with established neuroscience, establishing a powerful, evidence-based link between its interpretability and robust performance. All data, code, training logs and a one-command reproduction script are publicly available at `https://anonymous.4open.science/r/GACET-B6F8`.

## 1 Introduction

Brain state decoding with electroencephalography (EEG) plays a crucial role in brain–computer interfaces (BCIs) (Zhang et al., 2021). Continuous monitoring of spontaneous EEG signals enables inference of cognitive and affective states without explicit user intervention (Appriou et al., 2018; Al-Nafjan et al., 2017). This paradigm has found applications in stress detection (Sharma et al., 2022), fatigue monitoring (Othmani et al., 2023), sleep-stage classification (Wang et al., 2021), emotion recognition (Peng et al., 2022), and mental workload (MWL) assessment (Värbu et al., 2022). While a multitude of models have been developed to characterize dynamic brain states (Lotte et al., 2018), their practical utility remains severely limited by a critical, unresolved challenge.

That challenge is the profound variability of EEG signals across days (Yin & Zhang, 2017; Wu et al., 2022) and its even greater variability across subjects (Gibson et al., 2022) presents key obstacles for real-world deployment. Most studies still rely on single-day recordings due to the scarcity of large-scale, multi-day EEG datasets (Lin et al., 2017). As a result, decoding accuracy remains insufficient, and both cross-day and cross-subject generalization remain largely unverified (Wang et al., 2016; Kingphai & Moshfeghi, 2024). Systematically tackling this issue requires a two-pronged approach. First, a rigorous benchmark is needed to measure progress, addressing the scarcity of suitable multi-day EEG datasets. Second, a model architecture must be engineered for generalization to overcome the profound variability of EEG signals across days.

To address this foundational gap, we begin by introducing CD-MWL, the longest public MWL dataset to date. Comprising 42 h of EEG recordings from 14 participants over three days, it establishes a rigorous benchmark for safety-critical applications like aviation and adaptive training. Building upon this benchmark, we then propose GACET, a Graph-Aware Cross-domain EEG Transformer architecturally designed for generalization. GACET employs bidirectional cross-attention to fuse differential features, then dynamically learns a signal-dependent electrode graph via a Gumbel-

Softmax estimator followed by TransformerConv for topology-aware message passing. This end-to-end model achieves state-of-the-art (sota) accuracy and robustness in cross-day MWL recognition.

In summary, the main contributions of our paper are as follows:

1. **Largest Public EEG Dataset for MWL Assessment**: We present the largest publicly available EEG dataset named CD-MWL (cross days) for MWL evaluation. This dataset consists of recordings from 14 participants across three days and five difficulty levels, providing a comprehensive resource for future MWL research and model development.

2. **A Novel Interpretable Framework (GACET):** We introduce GACET, an interpretable framework that fuses spectral-temporal features using topology-aware graph reasoning. The model autonomously learns neurophysiologically plausible brain connectivity patterns, offering a transparent window into meaningful neural dynamics while achieving sota classification performance.

3. **A Rigorous, Transparent, and Reproducible Benchmark:** We establish a stringent benchmark to counteract common evaluation pitfalls like data leakage, ensuring fairness via exhaustive hyperparameter tuning for all baselines, and full transparency and reproducibility via publicly released code, training logs, and a one-click script.

## 2 RELATED WORK

### 2.1 PUBLIC EEG DATASETS FOR MWL

Table 1: Summary of Public EEG Datasets for MWL

| Dataset | Task | Level | CD | Channels | Sampling Rate | P*D | Duration |
|---|---|---|---|---|---|---|---|
| BVWM[2014] | VWM | 4 | No | 64 | 500 Hz | 13×1 | 3 hours |
| EHCD[2018] | N-BACK | 4 | No | 30 | 1000 Hz | 26×1 | 13 hours |
| STEW[2018] | SIMKAP | 4 | No | 14 | 128 Hz | 48×1 | 16.8 hours |
| EEGMAT[2019] | MA | 2 | No | 23 | 500 Hz | 36×1 | 2.4 hours |
| WM[2020] | VWM | 3 | **Yes** | 8 | 256 Hz | 9×(2-7)* | 6 hours |
| WAUC[2020] | MATB-II | 2 | No | 8 | 500 Hz | 48×1 | 16 hours |
| COG[2023] | MATB-II N-BACK | 4 | **Yes** | 64 | 500 Hz | 29×3 | 28 hours 32 hours |
| CL-Drive[2024] | VD | 10 | No | 4 | 256 Hz | 21×1 | 10.5 hours |
| CD-MWL | MATB-II | 5 | **Yes** | 64 | 1000 Hz | 14×3 | **42 hours** |

**Abbreviations:** CD = Cross Days, P*D = number of participants × number of days and * means that each participant performed the task over 2–7 separate days.

MWL is defined as the ratio of available resources to task demands (Wickens, 2008) and its paradigms fall into two categories: one cognitive-oriented, which assesses pure cognitive processing load (e.g., visual working memory (VWM) (Bashivan et al., 2014; Boran et al., 2020), mental arithmetic (MA) (Zyma et al., 2019) and n-back (Shin et al., 2018; Hinss et al., 2023)); and the other operation-oriented, which simulates real-world work scenarios and emphasizes multitasking and resource allocation (e.g., simultaneous capacity test (SIMKAP) (Lim et al., 2018), multi-attribute task battery-II (MATB-II) (Albuquerque et al., 2020; Hinss et al., 2023), vehicle driving (VD) (Angkan et al., 2024), and air traffic management (Aricò et al., 2015)).

Table 1 summarizes publicly available datasets, most of which are limited by low channel counts or single-day recordings, offering insufficient longitudinal data. This scarcity highlights the challenges in cross-day generalization research. For instance, while the valuable COG dataset offers multi-day recordings, its two tasks have non-equivalent difficulty scaling, which precludes their joint use for training a unified MWL model—the reason they are listed separately in our summary table. Its fixed task sequence may also introduce confounding fatigue effects. The WM dataset is unsuitable for our graph-based approach due to its low channel density and for robust personalized modeling due to its brief and highly variable per-subject recording times. Therefore, we selected the MATB-II task from

COG as an external benchmark alongside our newly collected CD-MWL, and further validated our model's cross-task generalization on the COG N-back task.

## 2.2 Deep learning methods for brain state decoding

Convolutional Neural Networks (CNNs) remain essential in EEG–BCI research, thanks to their capacity to extract spatial–temporal patterns directly from raw signals. Recent variants exemplify this progression: TSLANet (Eldele et al., 2024) incorporates an adaptive spectral block that merges FFT-based denoising with interaction convolutions to achieve robust performance on multi-channel recordings, and SFT-Net (Gao et al., 2024) combines depth-wise separable convolutions with frequency attention to facilitate lightweight fatigue detection.

Originally conceived for NLP and renowned for modeling long-range dependencies (Vaswani et al., 2017), Transformers have been increasingly applied to EEG decoding: MAET (Zhao & Gu, 2024) applies multi-head self-attention within an adaptive Transformer to compute attention weights over different view embeddings and fuse them via a weighted sum.

Some have combined both methods: MCA (Jiang et al., 2023) leverages cross-attention to fuse features, using bidirectional scaled dot-product attention within each band. The resulting fused 3D tensor is then passed into a customized and optimized 3D-CNN backbone.

Inspired by foundation models in vision and language, researchers have recently begun pre-training universal EEG backbones. EEGPT (Wang et al.) is trained on a diverse corpus to deliver task-agnostic representations that can be fine-tuned with minimal labeled data; similarly, LaBraM (Jiang et al., 2024) expands model capacity and capitalizes on roughly 2,500 hours of EEG signals to advance general-purpose feature learning.

## 3 Method

### 3.1 Dataset and preprocessing

**Dataset.** We utilized the MATB–II in this study, given its realistic multitasking simulation and adjustable difficulty levels. Originally developed at NASA's Langley Research Center (Hancock et al., 1995) and updated in 2011 as a modernized color version for Windows (Santiago-Espada et al., 2011) (for task details, refer to Appendix C), MATB–II serves as a robust benchmark for cognitive workload evaluation. To rigorously evaluate the performance of our model, we tested it on two EEG datasets featuring the MATB–II task: one publicly available and one collected in our laboratory.

Dataset 1 utilized a subset of the COG_BCI (Hinss et al., 2023). It included 29 participants, consisting of 11 females and 18 males, with a mean age of 23.9 ± 3.2 years. The participants completed three days one week apart. The participants performed the MATB–II task with four difficulty levels (with the resting state in the experiment designated as level 1). EEG data were recorded using a 64–channel electrode cap with the international 10-20 system at a sampling rate of 500 Hz.

Dataset 2 was collected in our laboratory as a dedicated MATB–II dataset. It comprised 14 participants, including 6 females and 8 males, aged 21–24 years. All experiments were run in a controlled laboratory environment, and sessions were scheduled during the daytime to minimize fatigue. Each participant completed three days at least 48 hours apart, with a 30-minute training session before the first. In each session, they performed the MATB–II task under five difficulty levels (for details of task difficulty design, see Appendix D), completing 3-minute blocks at each level with rest intervals between. Task order was counterbalanced using a Latin square design, shown in Figure 1, except that the resting level always appeared first. EEG data were recorded using a 64–channel electrode cap with the international 10-20 system at a sampling rate of 1000 Hz.

**Preprocessing.** EEG recordings contain environmental and physiological artifacts, so we preprocessed the data using the MNE-Python toolbox for cleaner and more reliable signals (Gramfort et al., 2013). A 1–100 Hz bandpass filter was first applied, followed by a 50 Hz notch filter to remove low-frequency drifts and powerline noise. Bad channels were interpolated, and signals were re-referenced to the average of all channels. The data were then downsampled to 500 Hz and independent component analysis (ICA) was performed for artifact removal. Finally, the data were

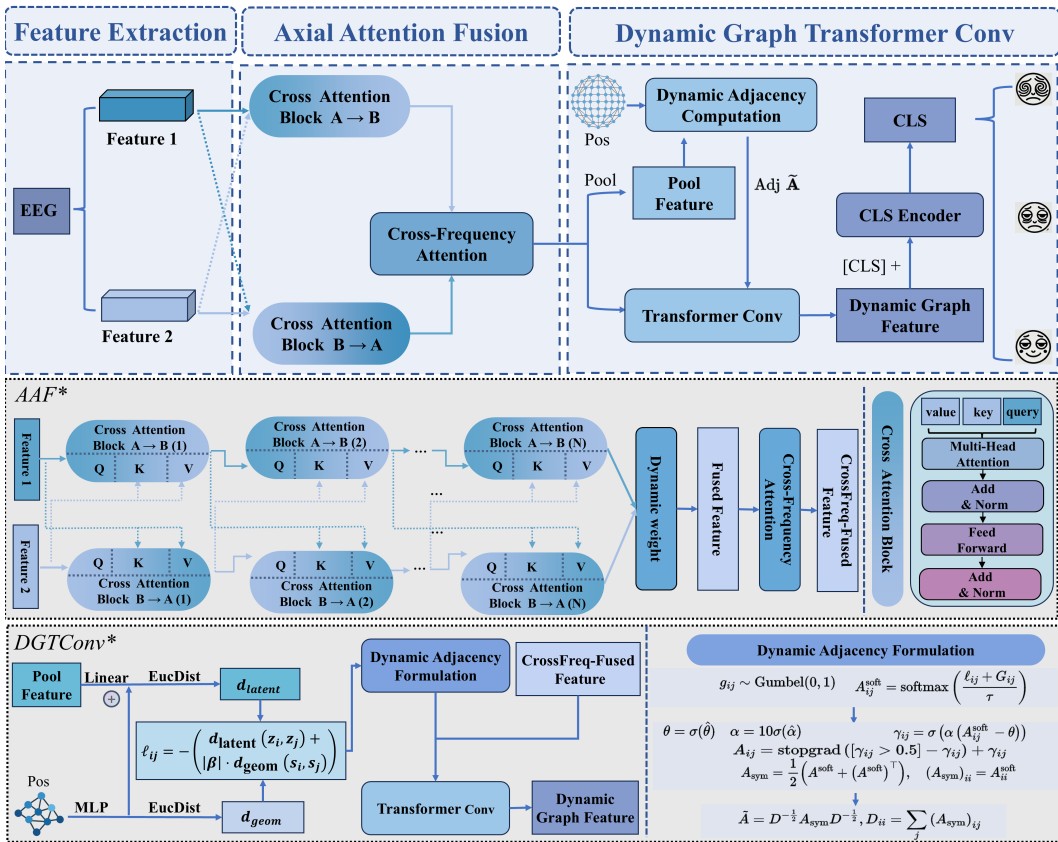

Figure 1: Experimental Workflow

Figure 2: **Overview of GACET architecture.** : (1) spectral–temporal Feature Extraction (DE & SampEn), (2) Axial Attention Fusion (AAF), and (3) Dynamic Graph Transformer Conv (DGTConv). Bottom: detailed views of the AAF module (denoted AAF*) on the left and the DGTConv module (denoted DGTConv*) on the right. *Legend:* Pool = pooling, Pos = electrode positions, EucDist = Euclidean distance.

segmented into non-overlapping, 16-second windows. For details of preprocessing parameters, please refer to Appendix E.

## 3.2 GACET

In this section we present **GACET**, a brain-state monitoring framework that takes MWL recognition as its entry point and delivers an end-to-end solution spanning spectral–temporal feature derivation, cross-domain representation learning and graph-aware spatial reasoning. The whole pipeline is depicted in Figure 2; it is organised into three functional blocks:

1. **Feature Extraction** (Figure 3): every 16-s pre-processed EEG segment $\mathbf{X} \in \mathbb{R}^{N \times N_c \times 8000}$ is band-pass filtered into Delta, Theta, Alpha, Beta, and Gamma bands. Differential Entropy (DE) and Sample Entropy (SampEn) are computed to form two token sequences that fully capture frequency-domain power and time-domain complexity.

2. **Axial Attention Fusion (AAF)**(Figure 2 *AAF\**): AAF first fuses DE and SampEn features using cross-attention blocks, then uses cross-frequency attention along the frequency axis to capture inter-band dependencies.

3. **Dynamic Graph Transformer Conv (DGTConv)** (Figure 2 *DGTConv\**): an electrode graph is sampled on-the-fly via a Gumbel-softmax hard-concrete scheme that blends geometric and latent distances. A shared TransformerConv propagates information through the graph at every time step.

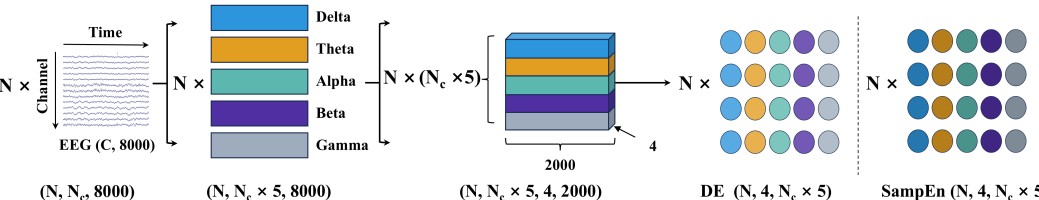

Figure 3: Feature extraction pipeline. $N$ is the number of samples, and $N_c$ is the number of channels.

**Feature extraction.** The preprocessed signals $X \in \mathbb{R}^{N \times N_c \times 8000}$, where $N$ and $N_c$ denote the number of samples and channels. With a uniform sampling rate of $500\,\mathrm{Hz}$ after preprocessing, each 16-second segment contains 8000 time points. Then we extract the Delta (1–4 Hz), Theta (4–8 Hz), Alpha (8–13 Hz), Beta (13–30 Hz), and Gamma (30–100 Hz) bands. The 8000-point sequence ($T = 8000$) is split into $S = 4$ non-overlapping temporal segments of length $L = T/S = 2000$, and differential entropy (DE) and sample entropy (SampEn) are computed for each segment.

*DE* is a nonlinear frequency-domain feature that demonstrates significant effectiveness in MWL evaluation (Wei et al., 2025). For a certain length of EEG signal that approximately obeys Gaussian distribution $\mathbb{N}(\mu, \sigma^2)$, its DE can be defined as (Li-Chen Shi et al., 2013):

$$h(X) = -\int_{-\infty}^{\infty} f(x) \log f(x) \mathrm{d}x = \frac{1}{2} \log \left( 2\pi e \sigma^2 \right) \tag{1}$$

For a discrete time segment of length $L$, the signal energy in the $i$-th frequency band can be estimated from the sample variance $\sigma_i^2$ as $P_i = L\sigma_i^2$, thus yielding:

$$h_i(X) = \frac{1}{2} \log \left( 2\pi e \sigma_i^2 \right) = \frac{1}{2} \log \left( L\sigma_i^2 \right) + \frac{1}{2} \log \frac{2\pi e}{L} = \frac{1}{2} \log \left( P_i \right) + \frac{1}{2} \log \frac{2\pi e}{L} \tag{2}$$

*SampEn* is a nonlinear time-domain measure that improves the approximate entropy by addressing its limitations (Richman & Moorman, 2000), and has been shown to be effective in MWL (Guan et al., 2021). To compute it, the time series is first reconstructed into template vectors of length $m$. We then count the number of vector pairs $(i, j)$ whose Chebyshev distance is within a tolerance $r$. Let $B$ be the total count of such matching pairs for vectors of length $m$, and $A$ be the total count for vectors of length $m + 1$. For a detailed derivation, please see Appendix F. The *SampEn* is then calculated as:

$$SampEn(m, r) = -\ln \frac{A}{B} \tag{3}$$

In practice, we use the algorithm's default parameters ($m = 2$ and $r = 0.2$) (Raphael, 2023).

**AAF.** The AAF module is designed to synergistically fuse two complementary EEG-derived feature sets: DE and SampEn. The input features, denoted as $X_{DE}$ and $X_{SampEn}$, are enriched with positional information by adding learnable positional embeddings, resulting in tensors of shape $(B, S, D)$, where $B$ is the batch size, $S$ is the sequence length, and $D$ is the feature dimension. The AAF module then operates on these feature representations in two primary stages.

The first stage focuses on facilitating bidirectional cross-domain interactions over several layers ($L_F$), enabling the DE and SampEn representations to mutually inform and refine each other. Within each layer $l \in [1, L_F]$ (with $X_{DE}^{(0)} = X_{DE}$ and $X_{SampEn}^{(0)} = X_{SampEn}$):

The DE features, $X_{DE}^{(l-1)}$, are updated by attending to the SampEn features, $X_{SampEn}^{(l-1)}$. Specifically, $X_{DE}^{(l-1)}$ acts as the Q, while $X_{SampEn}^{(l-1)}$ provides the K and V for a multi-head attention mechanism:

$$\text{Attn}_{DE \leftarrow SampEn}^{(l)} = \text{softmax}\left(\frac{X_{DE}^{(l-1)}W_Q\left(X_{SampEn}^{(l-1)}W_K\right)^\top}{\sqrt{d}}\right)X_{SampEn}^{(l-1)}W_V \quad (4)$$

The output of this attention operation is combined via a residual connection followed by layer normalization, and is then refined by a Feed-Forward Network (with residual connection and layer normalization), yielding $X_{DE}^{(l)}$. Similarly, applying the same process to $X_{SampEn}$ yields $X_{SampEn}^{(l)}$.

After $L_F$ such layers of iterative cross-domain refinement, the resulting advanced representations, $X_{DE}^{(L_F)}$ and $X_{SampEn}^{(L_F)}$, are integrated. This is achieved using a gated fusion mechanism. The two refined representations are first concatenated, and this combined tensor is passed through a small MLP to compute adaptive gating weights $w_a, w_b$ via a softmax function:

$$[w_a,\, w_b] = \text{softmax}\left(\text{reshape}_{(B,S,2,D)}\left(\text{MLP}_{\text{fusion}}\left([\, x_{DE}^{(L_F)},\, x_{SampEn}^{(L_F)}]\right)\right),\ \dim = 2\right) \quad (5)$$

These weights determine the contribution of each modality to the final fused representation, which is formed as a weighted sum, followed by a final layer normalization:

$$x_{\text{fused}} = \text{LayerNorm}(w_a \odot x_{DE}^{(L_F)} + w_b \odot x_{SampEn}^{(L_F)}) \quad (6)$$

In the second stage, the Cross-Frequency Attention module ingests the fused representation $X_{\text{fused}}$, whose embedding dimension $D$ is factorized into $N_C \times N_F$. T he tensor is then reshaped and permuted from $(B, S, D)$ to $(B, S, N_F, N_C)$ where $N_F$ correspond to the numbers of frequency bands, respectively. Axial self-attention is then applied along the frequency axis, where each band attends to every other, enabling the model to capture rich inter-band dependencies and integrate spectrum-wide contextual information. Through this, the network explicitly encodes interactions among frequency bands by leveraging multi-frequency features aggregated across all channels.

In summary, the AAF module yields a synergistic feature representation by bidirectionally refining the DE and SampEn streams and capturing spectrum-wide context, providing an optimized base for the subsequent DGTConv module can effectively perform topology-aware spatial reasoning.

**DGTConv.** The DGTConv module operationalizes our central hypothesis: that explicitly modeling the brain's dynamic network topology is key to robust generalization. It transforms static electrode coordinates into dynamic embeddings via positional projections plus context-driven offsets,computes Gumbel-Softmax–based adjacency scores by fusing latent and geometric distances, hardens them into a discrete adjacency matrix, and symmetrically normalizes it for graph convolution.

Initially, raw electrode positions $\mathbf{p}_i$ are centered, L2-normalized, and passed through a projection layer to produce basic positional embeddings $\mathbf{s}_i$. Concurrently, pooled features $\mathbf{g}$ are fed into a delta-MLP to generate dynamic offsets $\Delta_i(\mathbf{g})$. The final dynamic latent representation of each node is then the sum of these two components, $\mathbf{z}_i = \mathbf{s}_i + \Delta_i(\mathbf{g})$.

Based on these dynamic representations, we compute the similarity logits $\ell_{ij}$ between node pairs from a weighted combination of two distance metrics. The first is the latent-space distance, $d_{\text{latent}}(i, j) = \|\mathbf{z}_i - \mathbf{z}_j\|_2$, and the second is the geometric distance between the static embeddings, $d_{\text{geom}}(i, j) = \|\mathbf{s}_i - \mathbf{s}_j\|_2$. These are combined using a learnable scalar $\beta$ to form the final logit:

$$\ell_{ij} = -\big(d_{\text{latent}}(i, j) + |\beta|\, d_{\text{geom}}(i, j)\big). \quad (7)$$

To address the non-differentiability caused by discrete sampling when constructing the adjacency matrix, we draw on the Gumbel-Softmax method (Jang et al., 2017). Gumbel noise $G_{ij} \sim \text{Gumbel}(0, 1)$ is added to the similarity score $\ell_{ij}$, and the sum is divided by a learnable temperature parameter $\tau$.

Subsequently, soft adjacency probabilities $A_{ij}^{\text{soft}}$ are obtained via the Softmax function:

$$A_{ij}^{\text{soft}} = \text{softmax}_j \left( \frac{\ell_{ij} + G_{ij}}{\tau} \right) \tag{8}$$

where $\text{softmax}_j$ denotes normalization along the dimension of node $j$.

We extend the standard Gumbel–Softmax process by introducing learnable gating parameters $\alpha$ and $\theta$. First, given the soft adjacency value $A_{ij}^{\text{soft}}$ produced by Gumbel–Softmax, we compute

$$\gamma_{ij} = \sigma\big(\alpha\left(A_{ij}^{\text{soft}} - \theta\right)\big) \in (0, 1). \tag{9}$$

We then apply the straight-through estimator to "harden" $\gamma_{ij}$ into a binary decision:

$$A_{ij} = \text{stopgrad}\big([\gamma_{ij} > 0.5] - \gamma_{ij}\big) + \gamma_{ij}. \tag{10}$$

In the forward pass, $A_{ij} = [\gamma_{ij} > 0.5]$ yields a discrete 0/1 adjacency, while in the backward pass we enforce $\frac{\partial A_{ij}}{\partial \gamma_{ij}} = 1$ by treating the thresholding operation as the identity mapping. This ensures that we obtain a truly discrete adjacency matrix while preserving differentiability with respect to $\alpha$ and $\theta$.

To ensure symmetry and scale consistency, we average the discrete adjacency matrix $A$ with its transpose to obtain $A'_{\text{sym}} = \frac{1}{2}\left(A + A^T\right)$, and then replace the diagonal entries of $A'_{\text{sym}}$ with those of the soft adjacency matrix $A^{\text{soft}}$, i.e. $(A_{\text{sym}})_{ii} = A_{ii}^{\text{soft}}$, while off-diagonal entries remain $(A_{\text{sym}})_{ij} = (A'_{\text{sym}})_{ij}$ for $i \neq j$. Finally, we apply standard symmetric normalization to $A_{\text{sym}}$:

$$\widetilde{A} = D^{-\frac{1}{2}} A_{\text{sym}} D^{-\frac{1}{2}}, \quad D_{ii} = \sum_j (A_{\text{sym}})_{ij}. \tag{11}$$

Let the input be $X \in \mathbb{R}^{B \times S \times N_c \times N_f}$. We first apply LayerNorm to $X$. We then replicate the adjacency matrix $\widetilde{A} \in \mathbb{R}^{N_c \times N_c}$ along the $S$ time steps and merge all $S$ copies of each batch into a single graph with $BSN_c$ nodes. This batched graph is processed by a TransformerConv module, yielding an intermediate output of shape $(BSN_c) \times N_f$, which we reshape back to $B \times S \times D$. Finally, we apply a residual connection and a second LayerNorm to produce dynamic graph feature.

Finally, we prepend a `[CLS]` token to the dynamic graph features and feed them into a Transformer-based CLS encoder. We then extract the `[CLS]` representation for classification, with the depth of the classification head increasing alongside the number of classes, producing the final predictions.

## 4 EXPERIMENTS

### 4.1 IMPLEMENTATION DETAILS

Our experiments were conducted using Python 3.12.6, PyTorch 2.4.0, and CUDA 12.1 on two NVIDIA GeForce RTX 3090 GPUs. For both our model and the baselines, we employed the AdamW optimizer together with the standard cross-entropy loss function, ran a coarse grid search over 0.001, 0.0001, 0.00001, 0.000001 and used a fixed batch size of 32 throughout all experiments.

Because EEG signals are temporally autocorrelated, random shuffling before forming validation folds leaks adjacent trials into training and thus inflates non-cross-time (validation) accuracy (Shim et al., 2021). To prevent this, we used a two-stage split: first, leaving each recording day out in turn as an external test set (three splits); then, on the remaining two days, partitioning each subject's trials at each difficulty into five sequential, non-overlapping folds—ensuring that within each fold samples are temporally contiguous—yielding $3 \times 5 = 15$ runs per subject.

We set the random seed to 42 and standardized inputs using the training set's mean and variance (applied to validation and test). We applied early stopping—training was terminated if the 5-epoch average validation accuracy failed to improve by more than 0.01 for 10 epochs—and reduced the learning rate by 10× whenever the 5-epoch average validation accuracy didn't increase by over 0.01.

## 4.2 Performance Evaluation

We evaluated the model performance on Dataset 1 (binary classification between levels 1 and 4, and four-class classification across levels 1–4) and Dataset 2 (binary classification between levels 1 and 4, and five-class classification across levels 1–5). The results are presented in Table 2. All comparative models have undergone rigorous hyperparameter tuning to achieve their optimal performance. For more details, please refer to Appendix G.

Table 2: Comparison of different methods on Dataset 1 and Dataset 2 (cross-day).

| | Dataset 1 (2-class) | | Dataset 1 (4-class) | | Dataset 2 (2-class) | | Dataset 2 (5-class) | |
|---|---|---|---|---|---|---|---|---|
| | ACC (%) | F1 (%) | ACC (%) | F1 (%) | ACC (%) | F1 (%) | ACC (%) | F1 (%) |
| LaBraM[2024] | 60.37±4.17 | 41.17±6.41 | 32.77±5.06 | 20.33±6.13 | 71.31±8.83 | 67.30±11.65 | 28.34±5.44 | 18.02±6.07 |
| TSLANet[2024] | 71.45±12.01 | 68.67±13.43 | 45.93±6.51 | 44.30±6.58 | 75.79±7.98 | 74.47±8.79 | 29.63±3.21 | 27.92±3.39 |
| MAET[2023] | 96.27±3.17 | 96.15±3.29 | 65.97±9.18 | 62.84±10.10 | 89.69±4.84 | 89.06±5.86 | 40.67±4.63 | 37.00±5.73 |
| MCA[2024] | 91.75±5.37 | 91.21±6.01 | 58.82±8.15 | 56.70±9.21 | 84.64±5.49 | 83.85±6.00 | 35.81±3.34 | 32.27±3.98 |
| SFT-Net[2024] | 95.49±3.93 | 95.11±4.39 | 56.16±7.16 | 52.77±7.85 | 83.82±6.62 | 82.35±7.78 | 35.84±4.69 | 31.84±4.70 |
| **GACET** | **97.34±2.96** | **97.20±3.31** | **67.66±9.03** | **64.46±10.19** | **92.75±3.76** | **92.52±4.10** | **42.77±4.29** | **38.01±5.22** |

Table 3: Comparison of different methods on Dataset 1 and Dataset 2 (cross-subjects).

| | Dataset 1 (2-class) | | Dataset 1 (4-class) | | Dataset 2 (2-class) | | Dataset 2 (5-class) | |
|---|---|---|---|---|---|---|---|---|
| | ACC (%) | F1 (%) | ACC (%) | F1 (%) | ACC (%) | F1 (%) | ACC (%) | F1 (%) |
| LaBraM[2024] | 79.89±10.27 | 76.69±13.58 | 45.83±10.93 | 42.73±13.45 | 58.62±11.72 | 49.54±18.28 | 23.57±5.01 | 11.93±7.89 |
| TSLANet[2024] | 83.32±9.11 | 82.76±9.42 | 58.50±10.84 | 58.53±11.24 | 83.16±10.10 | **82.58±10.69** | 32.00±3.94 | 30.73±4.50 |
| MAET[2023] | 97.07±6.90 | 96.99±7.15 | 64.39±13.91 | 63.50±14.71 | 82.77±10.59 | 81.72±12.33 | 35.49±5.84 | 30.49±7.25 |
| MCA[2024] | 94.16±6.09 | 93.99±6.35 | 57.74±14.13 | 57.17±14.61 | 71.83±11.22 | 69.15±13.19 | 31.26±5.21 | 27.89±6.87 |
| SFT-Net[2024] | 86.85±14.03 | 84.36±19.03 | 54.33±7.73 | 52.04±10.18 | 64.68±16.12 | 56.59±22.16 | 32.27±8.09 | 26.45±10.73 |
| **GACET** | **97.54±5.44** | **97.48±5.60** | **65.25±14.57** | **64.37±15.72** | **83.36±12.74** | 81.79±15.91 | **36.21±6.84** | **31.44±9.72** |

In our primary cross-day evaluation (Table 2), GACET demonstrated clear superiority across all tasks on both datasets. For instance, it exceeded the next-best model by over 1.69 % on the four-class task of Dataset 1 and led by at least 3 % on the binary task of Dataset 2. To test for statistical significance, we fit a linear mixed-effects model for each comparison and confirmed that all of GACET's performance advantages are statistically significant, underscoring the reliability of its outperformance (see Appendix I). As an additional test of robustness, we evaluated the model in the more challenging leave-one-subject-out (LOSO) setting (Table 3), where GACET again achieved the highest accuracy on all tasks despite high inter-subject variability.

## 4.3 Generalization Analysis

Table 4: N-back task performance on Dataset 1.

| Model | 2-class | | 4-class | |
|---|---|---|---|---|
| | ACC (%) | F1 (%) | ACC (%) | F1 (%) |
| **GACET** | **99.94±0.14** | **99.94±0.15** | **58.35±11.49** | **57.61±12.33** |
| MAET | 99.71±0.63 | 99.70±0.66 | 57.08±10.71 | 56.05±11.58 |
| MCA | 93.14±3.88 | 92.87±4.02 | 47.24±6.09 | 47.91±6.58 |
| TSLANet | 94.45±2.92 | 93.58±3.65 | 45.28±3.45 | 44.76±4.74 |
| SFT_NET | 97.83±3.50 | 97.08±4.97 | 47.69±4.07 | 41.91±4.87 |
| LaBraM | 85.83±4.67 | 82.23±6.27 | 40.15±2.08 | 32.11±2.36 |

To validate our model's generalization capabilities, we benchmarked all models on the N-back task from Dataset 1 in a cross-time setting, following an identical evaluation protocol. As shown in Table 4, GACET again delivered the best performance with a statistically significant advantage over all baselines, confirming its robust generalization to a different cognitive workload paradigm.

## 4.4 Ablation study

As shown in Table 5, removing either of our core components individually degrades performance. The removal of DGTConv (M3) results in a moderate accuracy drop of approximately 2.7–4.1% across

Table 5: Ablation Study on Dataset 1 and Dataset 2. Methods: M1 = full model, M2 = w/o AAF, M3 = w/o DGTConv, M4 = w/o both modules.

| | Dataset 1 (2-class) | | Dataset 1 (4-class) | | Dataset 2 (2-class) | | Dataset 2 (5-class) | |
|---|---|---|---|---|---|---|---|---|
| | ACC (%) | F1 (%) | ACC (%) | F1 (%) | ACC (%) | F1 (%) | ACC (%) | F1 (%) |
| **M1** | **97.34±2.96** | **97.20±3.31** | **67.66±9.03** | **64.46±10.19** | **92.75±3.76** | **92.52±4.10** | **42.77±4.29** | **38.01±5.22** |
| M2 | 92.80±6.20 | 92.36±6.96 | 59.22±9.10 | 56.00±10.23 | 89.15±3.77 | 88.78±3.94 | 39.31±3.14 | 34.31±3.87 |
| M3 | 94.68±4.69 | 94.45±4.98 | 63.55±8.47 | 60.77±9.28 | 89.84±4.59 | 89.45±5.10 | 40.13±3.57 | 35.27±3.83 |
| M4 | 94.17±4.62 | 93.94±4.88 | 62.69±8.68 | 59.75±9.31 | 89.31±4.57 | 88.96±5.04 | 39.99±3.23 | 35.11±3.61 |

tasks. The effect is more pronounced for AAF; removing it (M2) causes a substantial performance drop, with accuracy decreasing by up to 8.44% on the four-class task of Dataset 1, confirming its critical role. Of particular importance is the powerful synergistic effect our analysis revealed between the AAF and DGTConv modules. The model lacking both components (M4) paradoxically outperforms the model lacking only AAF (M2) on several tasks (e.g., 94.17% vs. 92.80% ). This "module mismatch" strongly suggests that the DGTConv module's effectiveness is contingent upon receiving the refined, fused features produced by AAF. This finding moves beyond demonstrating the independent contribution of each module; it validates the holistic architectural design of GACET, where the components are not merely additive but mutually reinforcing.

## 4.5 NEUROPHYSIOLOGICAL ANALYSIS

Table 6: Node Degree Statistics by Region.

| Brain Region | Dataset 1 | Dataset 2 |
|---|---|---|
| | (Sum/Mean) | (Sum/Mean) |
| Parietal | 9.16 / 0.54 | 8.81 / 0.55 |
| Frontal | 7.41 / 0.46 | 6.88 / 0.49 |
| Central | 6.60 / 0.51 | 7.58 / 0.54 |
| Temporal | 3.19 / 0.35 | 1.35 / 0.22 |
| Occipital | 1.99 / 0.28 | 5.53 / 0.55 |

To validate our central hypothesis—that GACET achieves its superior performance by learning neurophysiologically meaningful brain dynamics—we analyzed the functional connectivity learned by the model. This was done by computing node degrees from the subject-averaged adjacency matrices of the binary classification tasks, followed by a 0-1 normalization. The structure represented by this matrix is not a predefined anatomical map but rather a dynamic representation of the brain's functional state. This representation is learned end-to-end, guided solely by classification loss, allowing the model to capture the inter-regional dynamics most salient for the cognitive task.

It reveals concentrated high connectivity in the Parietal, Frontal, and Central regions, a pattern consistent with the Fronto-parietal network's established role in managing cognitive workload (Marek & Dosenbach, 2018). Table 6 provides a quantitative summary, presenting both sum and mean node degrees to account for the differing number of electrodes per region. These statistics confirm this pattern and also highlight a key difference in the Occipital region. Given that MATB-II is a highly visual task, the substantially higher occipital connectivity in Dataset 2 (mean degree 0.55 vs. 0.28) likely reflects a greater visual load imposed by our different difficulty settings. By autonomously identifying a known functional network and sensitively capturing task-specific variations, GACET demonstrates that it is not merely fitting statistical patterns, but learning an effective and interpretable representation of the brain's cognitive workload dynamics.

## 5 CONCLUSION

To systematically address the critical challenge of cross-day generalization in BCI, this work delivers a unified solution. We first introduced CD-MWL, the largest public dataset for this purpose, establishing a rigorous benchmark for the community. On this benchmark, our proposed GACET model achieves state-of-the-art performance by learning neurophysiologically plausible brain connectivity. This powerful, evidence-based link between interpretability and robust performance validates our core hypothesis. By providing a complete package—a challenging dataset, a novel interpretable model, and a fully reproducible protocol—we establish a new standard for rigor and transparency, fostering progress towards real-world BCI applications.

## REPRODUCIBILITY STATEMENT

All code, our newly proposed CD-MWL dataset, and full training logs are publicly available at `https://anonymous.4open.science/r/GACET-B6F8`, including a one-click Jupyter Notebook script for easy verification of our core results. Our research package includes a fully programmatic preprocessing pipeline, which was applied to both the CD-MWL and the public COG datasets. To ensure fairness and prevent data leakage, we employed a rigorous two-stage cross-validation protocol and conducted exhaustive hyperparameter tuning for all baselines. Experiments were conducted using PyTorch 2.4.0 on two NVIDIA GeForce RTX 3090 GPUs, with a fixed random seed to ensure deterministic outcomes.

## ETHICS STATEMENT

The experimental protocol for the data collection in this study was reviewed and approved by the corresponding institutional review board. All participants provided written informed consent prior to their participation and received monetary compensation for their time. All collected data, including the publicly released CD-MWL dataset, have been fully anonymized to protect participant privacy. We foresee no direct negative societal impacts from this research.

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

## A  THE USE OF LARGE LANGUAGE MODELS

In the preparation of this manuscript, a LLM was utilized as an assistive tool. Its use was strictly limited to post-writing improvements of language and style for the manuscript text and standardization of formatting for the source code. The LLM was not used for any aspect of research ideation, data analysis, or the generation of substantive content.

## B  DETAILS OF PUBLIC EEG DATASET

**BVWM Dataset**

The BVWM dataset (Bashivan et al., 2014) comprises EEG recordings from 13 healthy participants performing a visual working memory task. Each participant completed 240 trials, where each trial lasted 3.5 seconds, including a 0.5-second character display and a 3-second maintenance period. In total, the dataset contains $13 \times 240 = 3,120$ trials, corresponding to a total task duration of 10,920 seconds (approximately 3.03 hours). Note that only the correctly responded trials (2,670) were used in model training and evaluation.EEG signals were recorded using 64 electrodes placed according to the international 10–10 system, sampled at 500 Hz. The cognitive load levels were defined by the number of characters to be memorized (2, 4, 6, or 8), corresponding to four workload conditions.

**EHCD Dataset**

The EEG–NIRS Hybrid Cognitive Dataset (EHCD) (Shin et al., 2018) comprises simultaneous EEG and near-infrared spectroscopy recordings from 26 right-handed healthy volunteers. EEG signals were captured using 30 active electrodes arranged according to the international 10–5 system and sampled at 1 000 Hz, while NIRS data were recorded via a NIRScout system with 16 sources and 16 detectors configured into 36 channels spanning frontal, motor, parietal, and occipital regions at 10.4 Hz .

Participants completed three cognitive paradigms in separate sessions. The n-back task comprised four difficulty levels (rest, 0-, 2-, and 3-back); each session consisted of nine series of 2 s instruction, 40 s of digit presentations (one every 2 s), and 20 s rest—totaling roughly 27 minutes per participant . The DSR paradigm featured two difficulty conditions (target vs. non-target) and followed the identical timing for about 27 minutes per participant . In the Word Generation paradigm, two conditions (word generation vs. baseline) were presented in 20 trials per session—each trial including a 2 s cue, 10 s task, and 13–15 s rest—also amounting to approximately 27 minutes of recording per participant . With 26 participants, each paradigm yields on the order of 13 hours of combined EEG and NIRS data across all participants.

**STEW Dataset**

The STEW dataset (Lim et al., 2018) comprises continuous EEG recordings from 48 graduate participants, acquired with a 14-channel Emotiv EPOC headset (10–20 system) at 128 Hz . Each session began with a 3-minute eyes-open resting-state baseline (one condition), followed by an 18-minute Simultaneous Capacity Test (SIMKAP). For workload analysis, the final 3 minutes of SIMKAP were selected and trimmed by 15 seconds at both the start and end—yielding a 2.5-minute "workload" segment . Participants rated their mental workload on a 1–9 scale after each segment; these subjective scores were then binned into three levels—low (1–3), moderate (4–6), and high (7–9)—resulting in four conditions (rest plus three workload levels) per participant.

**EEGMAT Dataset**

The EEGMAT dataset (Zyma et al., 2019) contains EEG recordings from 36 healthy volunteers. Data were acquired with a 23-channel Neurocom monopolar EEG system (International 10/20 montage) at a 500 Hz sampling rate. Each subject contributed artifact-free EEG segments of 180 s during eyes-closed resting and 60 s during a continuous mental arithmetic task (serial subtraction of two-digit numbers from four-digit numbers).

**WM dataset** The WM dataset (Boran et al., 2020) includes EEG recordings from 9 epilepsy patients performing a verbal working memory task. Each subject completed multiple sessions (total 37), with 50 trials per session, yielding a total task duration of approximately 370 minutes (6.17 hours).EEG

was recorded using the 10–20 system at 256 Hz. Each trial comprised a 2-second encoding phase, a 3-second maintenance period, and a response stage.

**WAUC Dataset**

The WAUC dataset (Albuquerque et al., 2020) includes recordings from 48 participants. It captures seven synchronized modalities: EEG, ECG, respiration rate, skin temperature, galvanic skin response, blood volume pulse, and three-axis accelerometry. EEG data were acquired with 8 channels at a 500 Hz sampling rate. Participants performed the MATB tasks under two levels of mental workload (low vs. high) and three levels of physical activity. Each combined mental-physical task block lasted approximately 10 minutes .

**COG-BCI Dataset**

The COG-BCI dataset (Hinss et al., 2023) comprises EEG recordings from 29 healthy volunteers collected over three sessions spaced one week apart. Signals were acquired with a 64-channel ActiCap cap following the international 10–20 layout, referenced at Fpz, and sampled at 500 Hz .

In each session, participants performed the MATB-II paradigm under four conditions — rest (4 min) and easy, medium, and difficult runs of approximately 5 min each ($\approx$ 19 min total); the N-back paradigm under four conditions —rest (4 min) and 0-, 1-, and 2-back blocks of approximately 6 min each ($\approx$ 22 min total); the Psychomotor Vigilance Task lasting about 10 min ; and the Eriksen Flanker task lasting about 10 min.

**CL-Drive Dataset**

The CL-Drive dataset (Angkan et al., 2024) comprises recordings from 21 participants, capturing four synchronized modalities: EEG, ECG, electrodermal activity (EDA), and eye-tracking . EEG was acquired via a Muse S headband with four channels (AF7, AF8, TP9, TP10) at a 256 Hz sampling rate . Participants completed nine driving scenarios of graded complexity—and an interleaved resting baseline—yielding a total of 10 difficulty conditions. Each scenario or rest block lasted three minutes, separated by two-minute intervals, and during each driving block they self-reported cognitive load every 10 seconds. Overall, each participant contributed approximately 45 minutes of recorded data, including both task and baseline periods.

## C  MATB

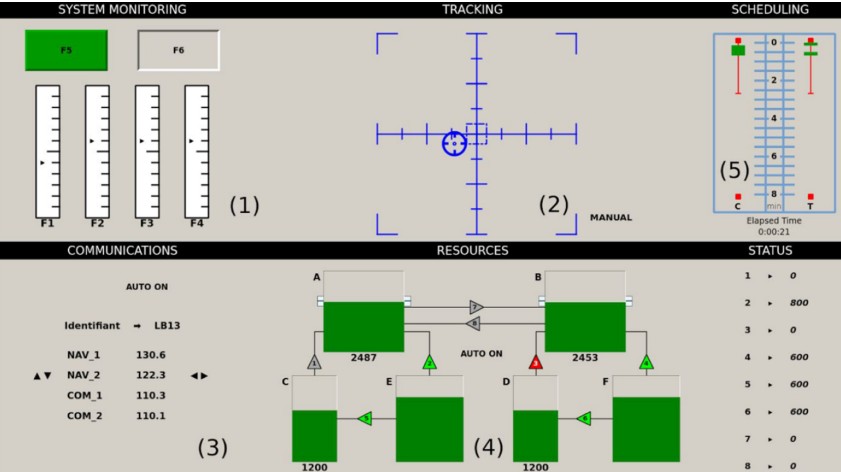

Figure 4: MATB

This experiment simulates a multitasking environment to assess participants' performance under high workload conditions. The tasks are designed to reflect typical activities encountered in piloting scenarios, including system monitoring, dynamic target control, communication handling, and resource allocation. Participants are required to manage multiple modules simultaneously, switching their attention effectively within a limited timeframe. The interface is organized into modular sections, and interactions are facilitated through a keyboard and joystick. Performance is measured using

metrics such as accuracy, response time, and task stability. Additionally, a timeline view assists participants in planning task sequences, providing a realistic representation of decision-making and mental workload in complex operational contexts (Cegarra et al., 2020).

1.The System Monitoring Task

Participants monitor multiple indicators on the screen and respond promptly to abnormalities. Two light indicators (F5 and F6) at the top have default states: one green and one gray. When their states change (e.g., the green light turns off or the gray light turns red), participants must press the corresponding keys (F5 or F6) to restore them. Additionally, four gauges (F1 to F4) at the bottom display pointers that are normally centered. If a pointer moves outside its central range, participants need to recenter it by pressing the corresponding key. Actions must be completed within a set timeframe; delays result in task failure.

2.The Tracking Task

Participants use a joystick to control a cursor on the screen, keeping it within a designated target area. The cursor's movement follows a sinusoidal path, which is complex but predetermined. Participants must make precise adjustments to avoid the cursor leaving the target area. Performance is assessed by calculating the RMS deviation of the cursor from the target area.

3.The Communication Task

Participants adjust communication channels and frequencies based on auditory instructions that include their call sign (e.g., "GB54"). Instructions may specify actions such as "Adjust COM1 to frequency 130.5." Unrelated instructions must be ignored. Participants use the up/down arrow keys to select the channel, the left/right keys to adjust the frequency, and the Enter key to confirm their actions. Visual markers on the screen help participants verify their adjustments.

4.The Resource Management Task

Participants manage a fuel system consisting of six tanks, two of which deplete fuel at a constant rate. They must operate eight pumps to redistribute fuel and maintain critical tank levels within a safe range. Some supply tanks have capacity limits (e.g., 2000 units), while others have unlimited capacity. Pump flow rates vary, with high-rate pumps transferring 800 units per minute and others operating at 600 or 400 units. Participants actively control pump states to balance fuel levels. Performance is measured by the time tank levels remain within the target range or by calculating RMS error.

5.The Scheduling View

The scheduling view, located at the top-right of the screen, provides a timeline of task events over the next several minutes. Red and green markers indicate task statuses, such as inactive periods (red) and active periods requiring action (green). A digital timer shows the elapsed time since the experiment began. While participants do not interact directly with the scheduling view, it aids in anticipating task changes and optimizing task prioritization.

# D  DETAILS OF DIFFICULTY DESIGN

Table 7: Task Parameter Settings Across Different Difficulty Levels

| | System Monitoring Task | | Tracking Task | | Communication Task | | Resource Management Task | |
|---|---|---|---|---|---|---|---|---|
| Difficulty | Event Interval (s) | Interval Variability (s) | Joystick Sensitivity | Light Motion Speed | Event Interval (s) | Call Frequency (rate) | Failure Interval (s) | Failure Duration (s) |
| Level 1 | Nan | Nan | Nan | Nan | Nan | Nan | Nan | Nan |
| Level 2 | 30 | 5 | 0.5 | 0.06 | 40 | 0.5 | 40 | 10 |
| Level 3 | 20 | 5 | 2.0 | 0.1 | 30 | 0.5 | 20 | 10 |
| Level 4 | 10 | 2 | 5.0 | 0.45 | 20 | 0.8 | 5 | 15 |
| Level 5 | 5 | 2 | 10.0 | 0.8 | 10 | 0.8 | 5 | 15 |

In Table 7, the task parameter settings across different difficulty levels are summarized.In Level 1 (resting state), no task is performed. Therefore, all task-related parameters are marked as 'NaN' to indicate no applicable values.

For Levels 2 to 5, the parameter settings are defined as follows:

The System Monitoring Task: The event interval and its variability decrease as the difficulty level increases, making the system monitoring tasks more frequent and less predictable.

The Tracking Task: Joystick sensitivity decreases (a larger value corresponds to lower sensitivity), and the speed of the moving light increases, making precise control more difficult.

The Communication Task: The event interval shortens and the call frequency increases at higher difficulty levels, requiring faster response times.

The Resource Management Task: The failure interval shortens, while the failure duration remains constant or slightly increases, introducing more frequent and prolonged management challenges at higher difficulty levels.

This design ensures that higher difficulty levels impose increasingly complex and demanding cognitive loads, simulating real-world multitasking scenarios.

## E    DETAILS OF THE EEG PREPROCESSING

**Channel exclusion:**

`raw.drop_channels(['Cz'])` (Dataset 1 only; Cz was not recorded for participants 1–9).

`raw.drop_channels(['CB1','CB2','HEO','VEO','EMG','EKG','M1','M2'])` (Dataset 2 only; CB1/CB2 are non-standard, M1/M2 unused for referencing, others are non-EEG signals).

**Filtering:**

`raw.filter(l_freq=1, h_freq=100, picks='eeg');`

`raw.notch_filter(freqs=(48, 52), picks='eeg').`

**Bad-channel detection & interpolation:**

`bads = mne.preprocessing.find_bad_channels_lof(raw, threshold=2.0);`

**Re-referencing:**

`raw.set_eeg_reference(ref_channels='average', ch_type='eeg').`

**Down-sampling:**

`raw.resample(sfreq=500)` (Dataset 2 only).

**ICA artifact removal:**

- Fit ICA:
  `ica = mne.preprocessing.ICA(method='infomax', extended=True, max_iter='auto', random_state=42);`
- Muscle artifacts:
  `ica.find_bads_muscle(raw, threshold=0.9).`
- ECG artifacts (Dataset 1 only, as only Dataset 1 includes ECG recordings):
  `ica.find_bads_ecg()`
- Eye-blink artifacts:
  `ICLabel probability > 0.9`
- Exclude & apply:
  `ica.exclude = muscle_idx + ecg_idx + blink_idx;`

## F    DETAILED DERIVATION OF SAMPLE ENTROPY (*SampEn*)

The calculation of *SampEn* for a scalar time series $\{x_1, x_2, \ldots, x_L\}$ of length $L$ with embedding dimension $m$ and tolerance $r$ proceeds as follows:

**Form Template Vectors**    The $m$- and $(m+1)$-dimensional template vectors are formed:

$$X_m(i) = [x_i,\ x_{i+1},\ \ldots,\ x_{i+m-1}],\quad i = 1, 2, \ldots, L - m + 1;$$
$$X_{m+1}(i) = [x_i,\ x_{i+1},\ \ldots,\ x_{i+m}],\quad i = 1, 2, \ldots, L - m. \tag{12}$$

**Measure Distance**   Similarity between any two vectors $u$ and $v$ is measured by the Chebyshev distance:

$$d_\infty(u, v) = \max_{0 \le k < \dim(u)} \big| u_k - v_k \big|. \tag{13}$$

**Count Pairs**   All pairs whose Chebyshev distance does not exceed $r$ are counted and denoted as:

$$\begin{aligned} B &= \#\{(i,j) : i \ne j,\ d_\infty(X_m(i), X_m(j)) \le r\}, \\ A &= \#\{(i,j) : i \ne j,\ d_\infty(X_{m+1}(i), X_{m+1}(j)) \le r\}. \end{aligned} \tag{14}$$

**Define *SampEn***   Finally, the *SampEn* is defined as shown in Equation 3:

$$\mathrm{SampEn}(m, r) \; = \; -\ln\!\Big(\tfrac{A}{B}\Big). \tag{15}$$

## G   OVERALL

This section presents the experimental results of all comparative models under different learning rates. All models use the input tensor of shape $(N, C, 8000)$ as illustrated in Figure 3 and keep $N$ fixed.

For LaBraM, we adopted four approaches: **arch_train** uses only the network architecture and trains from scratch on inputs of shape $(N, C, 8000)$; **eval_only** loads the pre-trained weights and evaluates without further training; **finetune_full** loads the pre-trained weights and fine-tunes all parameters; and **finetune_head** loads the pre-trained weights and fine-tunes only the classification head. We perform a coarse search over the learning rate set (0.001, 0.0001, 0.00001, 0.000001). All training logs and detailed results are available in the `log` directory of the code.

Power Spectral Density (PSD) represents the distribution of power across different frequencies, which has been used in MCA (Jiang et al., 2023).Since we computed PSD, we also evaluated its performance as an input feature in other models. We employ a 512-point Short-Time Fourier Transform (STFT) with the non-overlapping Hanning window to compute PSD. The specific calculation process is as follows.

For a given one-dimensional signal $x(n)$ which is divided into $K$ segments, each of length $L$. The $i$–th segment is denoted as $x_i(n) = x(1), x(2), \ldots, x(L)$, for $i = 0, 1, \ldots, K - 1$.

The periodogram for the $i$–th segment is calculated using the window function $w(n)$ as follows:

$$\hat{p}_i(f) = \frac{1}{\sum_{n=0}^{L-1} |w(n)|^2} \sum_{n=0}^{L-1} \big| w(n) x_i(n) e^{-j2\pi f n} \big|^2 \tag{16}$$

Here, $f$ denotes the frequency, and $j$ is the imaginary unit. The Welch-PSD is then obtained by averaging the periodograms of all $K$ segments:

$$\hat{p}_w(f) = \frac{1}{K} \sum_{i=0}^{K-1} \hat{p}_i(f) \tag{17}$$

### G.1   LABRAM

For data utilizing pretrained weights, we followed the preprocessing protocol from the original work (Jiang et al., 2024): applying a $0.1$–$75$ Hz band-pass filter, a $50$ Hz notch filter, downsampling to $200$ Hz, and normalizing to the $\pm 1$ range.

### G.2   TSLANET

TSLANet (Eldele et al., 2024) employs an adaptive spectral block (ASB), utilizing FFT-based thresholding denoising to learn both long- and short-term dependencies; subsequently, an interaction convolution block is applied to deepen spatiotemporal coupling. Since the original design uses a limited number of physiological signal channels (two EEG channels and one ECG channel), we

Table 8: LaBraM_overall

| | Dataset 1 (2-class) | | Dataset 1 (4-class) | | Dataset 2 (2-class) | | Dataset 2 (5-class) | |
|---|---|---|---|---|---|---|---|---|
| | ACC (%) | F1 (%) | ACC (%) | F1 (%) | ACC (%) | F1 (%) | ACC (%) | F1 (%) |
| 1e-3 | | | | | | | | |
| eval_only | 49.50±1.31 | 36.34±1.80 | 24.35±0.89 | 12.68±0.81 | 49.97±0.83 | 36.97±1.54 | 19.86±0.30 | 8.21±0.54 |
| LaBraM_arch_train | 63.82±3.13 | 41.16±5.34 | 28.97±2.43 | 14.91±3.27 | 52.52±1.82 | 36.75±2.61 | 29.68±3.90 | 20.83±4.58 |
| LaBraM_finetune_full | 55.99±4.13 | 43.49±5.81 | 29.30±4.13 | 19.91±4.92 | 57.43±8.69 | 46.10±12.11 | 23.45±4.86 | 12.06±6.21 |
| LaBraM_finetune_head | 55.22±4.39 | 44.56±6.14 | 27.95±2.77 | 21.68±2.93 | 61.17±8.79 | 58.26±10.11 | 24.57±3.81 | 20.36±3.80 |
| 1e-4 | | | | | | | | |
| **LaBraM_arch_train** | **60.37±4.17** | **41.17±6.41** | **32.77±5.06** | **20.33±6.13** | **71.31±8.83** | **67.30±11.65** | **28.34±5.44** | **18.02±6.07** |
| LaBraM_finetune_full | 56.91±4.33 | 44.80±6.93 | 31.22±5.37 | 24.26±6.03 | 67.98±10.48 | 64.81±12.84 | 27.48±6.11 | 20.33±7.03 |
| LaBraM_finetune_head | 55.70±0.52 | 35.79±0.22 | 26.37±1.19 | 13.30±1.43 | 55.11±6.84 | 45.85±10.01 | 22.99±3.55 | 15.62±3.44 |
| 1e-5 | | | | | | | | |
| LaBraM_arch_train | 62.50±0.00 | 38.46±0.00 | 34.33±4.83 | 21.29±6.22 | 67.51±9.86 | 62.62±13.10 | 29.16±4.46 | 19.60±3.84 |
| LaBraM_finetune_full | 56.22±1.03 | 37.48±1.83 | 27.62±2.20 | 15.58±2.47 | 60.46±9.57 | 54.74±12.35 | 24.90±5.37 | 15.90±5.42 |
| LaBraM_finetune_head | 55.70±0.53 | 35.77±0.22 | 26.22±0.60 | 11.92±0.99 | 52.90±3.84 | 40.49±5.65 | 22.14±2.91 | 12.57±3.04 |
| 1e-6 | | | | | | | | |
| LaBraM_arch_train | 62.16±1.49 | 39.63±2.09 | 28.21±1.42 | 13.11±1.42 | 56.08±4.94 | 45.16±7.15 | 24.98±3.28 | 15.08±3.38 |
| LaBraM_finetune_full | 54.54±1.92 | 38.43±1.85 | 27.10±1.69 | 15.76±1.67 | 56.26±6.55 | 47.81±8.96 | 21.61±1.90 | 10.62±2.25 |
| LaBraM_finetune_head | 55.70±0.53 | 35.77±0.22 | 26.36±1.03 | 12.33±1.48 | 52.45±3.44 | 39.80±5.08 | 21.71±2.89 | 11.56±3.09 |

explored extending its network depth. Here, TSLANet $_n$ denotes the variant in which both the embedding dimension and the model depth are scaled by a factor of $n$ (with $n = 1$ corresponding to the original model).

## G.3 MAET

MAET (Zhao & Gu, 2024) (Multi-view Embedding Module + Adaptive Transformer) utilizes differential entropy (DE) as the input feature; additionally, we evaluated power spectral density (PSD) and sample entropy (SampEn), and ultimately selected DE with a learning rate of $5 \times 10^{-4}$ as yielding the best performance.

## G.4 MCA

MCA (Jiang et al., 2023) employs DE and PSD as dual features, fusing them via cross attention mechanisms before classification with a customized 3D convolutional neural network.In addition to DE + PSD, we also explored the results of DE + SampEn and PSD + SampEn.

## G.5 SFT-NET

SFT(Gao et al., 2024) employs spatial and frequency attention mechanisms, followed by depthwise separable convolution with contextual structures, and a two-layer LSTM—thereby balancing inter-preta bility and lightweight design . In the original model, each sample requires 16 DE segments. Through our feature computation, each sample contains only 4 segments. Since no interface was provided, we modified the code accordingly, naming the variant SFT$_4$. We also computed SFT$_{16}$ using a 1 s window—keeping the total number of segments per sample constant—to adapt the model. Additionally, we evaluated the network's performance when using PSD and SampEn as input features.

## G.6 GACET

Our model also explored strategies beyond the combination of DE and SampEn, but the results showed that the combination of DE and SampEn still achieved the best performance, confirming the validity of our feature selection.

Table 9: TSLANet_overall

| | Dataset 1 (2-class) | | Dataset 1 (4-class) | | Dataset 2 (2-class) | | Dataset 2 (5-class) | |
| --- | --- | --- | --- | --- | --- | --- | --- | --- |
| | ACC (%) | F1 (%) | ACC (%) | F1 (%) | ACC (%) | F1 (%) | ACC (%) | F1 (%) |
| **1e-3** | | | | | | | | |
| TSLANet_1 | 53.38±5.43 | 51.33±5.53 | 29.57±3.71 | 28.48±3.82 | 62.72±5.52 | 62.16±5.38 | 25.69±2.51 | 25.11±2.52 |
| TSLANet_2 | 58.85±7.72 | 55.42±8.61 | 33.69±4.28 | 32.66±4.69 | 68.57±6.49 | 67.84±6.71 | 26.99±3.02 | 26.18±2.99 |
| TSLANet_3 | 62.97±10.79 | 59.95±11.92 | 37.12±5.48 | 35.76±5.78 | 70.74±6.85 | 69.81±7.33 | 27.81±2.76 | 26.85±2.86 |
| TSLANet_4 | 67.79±12.65 | 65.38±13.58 | 40.53±6.01 | 39.16±6.17 | 72.99±6.92 | 72.01±7.41 | 28.48±2.79 | 27.39±2.96 |
| TSLANet_5 | 70.14±12.60 | 67.60±14.04 | 42.34±6.14 | 40.91±6.50 | 73.59±6.84 | 72.64±7.27 | 28.65±2.84 | 27.32±3.05 |
| TSLANet_6 | 70.36±13.34 | 67.71±14.91 | 44.23±6.47 | 42.85±6.42 | 74.50±7.68 | 73.36±8.49 | 29.27±3.17 | 27.93±3.32 |
| TSLANet_7 | 71.84±12.43 | 69.22±13.93 | 44.73±6.22 | 43.05±6.38 | 75.76±7.00 | 74.73±7.63 | 29.02±2.91 | 27.52±2.94 |
| **TSLANet_8** | **71.45±12.01** | **68.67±13.43** | **45.93±6.51** | **44.30±6.58** | **75.79±7.98** | **74.47±8.79** | **29.63±3.21** | **27.92±3.39** |
| **1e-4** | | | | | | | | |
| TSLANet_1 | 55.71±7.01 | 53.24±7.12 | 31.38±4.09 | 30.56±4.33 | 63.93±5.46 | 63.46±5.35 | 24.66±2.56 | 24.37±2.55 |
| TSLANet_2 | 60.91±8.76 | 57.99±9.33 | 35.80±4.95 | 35.04±5.18 | 69.95±5.75 | 69.30±5.95 | 27.46±3.16 | 26.76±3.19 |
| TSLANet_3 | 62.84±10.00 | 59.79±10.74 | 38.77±5.83 | 38.19±6.03 | 71.66±6.32 | 70.91±6.75 | 28.92±3.17 | 27.90±3.16 |
| TSLANet_4 | 63.75±9.86 | 61.03±10.45 | 40.22±6.35 | 39.70±6.48 | 72.47±6.75 | 71.76±7.10 | 29.40±3.49 | 28.22±3.41 |
| TSLANet_5 | 64.27±10.75 | 61.55±11.48 | 41.78±6.66 | 41.14±6.77 | 72.72±6.53 | 71.92±7.09 | 29.88±3.24 | 28.62±3.21 |
| TSLANet_6 | 64.47±11.25 | 62.07±11.72 | 42.21±7.07 | 41.56±7.15 | 73.32±6.08 | 72.58±6.50 | 30.37±3.46 | 28.99±3.40 |
| TSLANet_7 | 65.16±11.46 | 62.75±12.27 | 42.47±7.26 | 41.58±7.37 | 73.64±6.75 | 72.82±7.26 | 30.66±3.51 | 29.25±3.46 |
| TSLANet_8 | 66.60±11.71 | 64.32±12.56 | 43.21±6.77 | 42.28±6.83 | 74.38±6.85 | 73.60±7.25 | 30.91±3.56 | 29.49±3.50 |
| **1e-5** | | | | | | | | |
| TSLANet_1 | 58.00±7.45 | 55.78±7.66 | 31.54±4.64 | 30.66±4.81 | 61.29±4.76 | 60.55±4.77 | 23.01±2.01 | 22.61±1.98 |
| TSLANet_2 | 62.56±9.46 | 60.29±9.84 | 35.15±5.12 | 34.50±5.53 | 68.56±6.45 | 67.80±6.60 | 25.93±2.41 | 25.50±2.57 |
| TSLANet_3 | 65.21±10.41 | 62.92±11.00 | 38.24±6.16 | 37.86±6.56 | 73.00±7.10 | 72.27±7.42 | 27.92±3.18 | 27.33±3.30 |
| TSLANet_4 | 66.80±11.44 | 64.54±12.22 | 40.16±6.53 | 39.87±6.96 | 74.46±7.23 | 73.72±7.66 | 29.05±3.34 | 28.27±3.39 |
| TSLANet_5 | 67.68±11.77 | 65.42±12.65 | 41.32±6.74 | 41.07±7.01 | 75.20±7.23 | 74.47±7.67 | 29.80±3.35 | 28.83±3.46 |
| TSLANet_6 | 68.39±11.68 | 66.24±12.61 | 41.75±7.04 | 41.40±7.31 | 75.63±7.52 | 74.92±7.93 | 30.54±3.47 | 29.44±3.43 |
| TSLANet_7 | 68.61±11.91 | 66.41±12.79 | 42.55±6.99 | 42.24±7.25 | 75.98±7.53 | 75.28±7.98 | 31.03±3.83 | 29.76±3.78 |
| TSLANet_8 | 69.34±11.71 | 67.09±12.85 | 42.67±7.01 | 42.31±7.32 | 76.39±7.59 | 75.70±7.95 | 31.13±3.78 | 29.76±3.60 |
| **1e-6** | | | | | | | | |
| TSLANet_1 | 54.42±4.74 | 53.09±4.73 | 28.41±3.51 | 27.50±3.53 | 57.18±4.45 | 56.19±4.52 | 22.07±1.96 | 21.10±1.99 |
| TSLANet_2 | 57.98±6.62 | 56.39±6.73 | 30.99±4.32 | 29.83±4.55 | 59.45±4.22 | 57.87±4.41 | 23.19±1.85 | 22.28±1.97 |
| TSLANet_3 | 59.38±7.37 | 57.70±7.45 | 32.19±4.89 | 31.13±5.47 | 62.17±5.28 | 60.37±5.40 | 24.00±2.10 | 22.85±2.23 |
| TSLANet_4 | 60.78±8.71 | 59.47±8.86 | 33.40±5.46 | 32.35±5.95 | 63.23±5.33 | 61.23±5.66 | 24.49±2.25 | 23.42±2.33 |
| TSLANet_5 | 62.20±8.65 | 60.39±8.88 | 34.44±5.26 | 33.51±5.70 | 65.30±5.21 | 63.50±5.45 | 25.27±2.22 | 24.13±2.39 |
| TSLANet_6 | 63.24±9.78 | 61.45±10.10 | 35.05±5.83 | 34.16±6.44 | 66.58±5.32 | 64.90±5.33 | 25.98±2.40 | 24.84±2.41 |
| TSLANet_7 | 62.93±9.80 | 61.26±10.10 | 35.85±5.86 | 35.06±6.39 | 67.00±5.94 | 65.39±6.18 | 26.65±2.58 | 25.50±2.65 |
| TSLANet_8 | 63.47±10.13 | 61.86±10.46 | 36.10±6.01 | 35.16±6.65 | 68.42±6.14 | 67.03±6.34 | 26.85±2.74 | 25.72±2.81 |

Table 10: MAET_overall

| | Dataset 1 (2-class) | | Dataset 1 (4-class) | | Dataset 2 (2-class) | | Dataset 2 (5-class) | |
|---|---|---|---|---|---|---|---|---|
| | ACC (%) | F1 (%) | ACC (%) | F1 (%) | ACC (%) | F1 (%) | ACC (%) | F1 (%) |
| **1e-3** | | | | | | | | |
| **MAET_single_DE** | **96.27±3.17** | **96.15±3.29** | **65.97±9.18** | **62.84±10.10** | **89.69±4.84** | **89.06±5.86** | **40.67±4.63** | **37.00±5.73** |
| MAET_single_PSD | 95.88±2.69 | 95.69±2.85 | 62.96±9.00 | 60.91±9.57 | 86.09±6.23 | 85.49±7.12 | 37.22±3.56 | 34.42±3.89 |
| MAET_single_SampEn | 92.22±5.56 | 91.82±6.24 | 55.39±6.96 | 54.17±7.63 | 86.50±5.13 | 86.26±5.34 | 38.78±3.53 | 37.41±3.65 |
| **1e-4** | | | | | | | | |
| MAET_single_DE | 95.98±3.45 | 95.83±3.60 | 64.68±8.11 | 61.98±9.37 | 88.87±4.54 | 88.32±5.33 | 41.10±4.30 | 36.58±5.03 |
| MAET_single_PSD | 94.65±3.47 | 94.46±3.62 | 61.62±8.06 | 59.76±8.94 | 85.88±5.68 | 85.36±6.30 | 37.71±3.57 | 34.36±3.86 |
| MAET_single_SampEn | 88.12±6.50 | 87.61±7.06 | 51.63±6.28 | 49.84±7.01 | 84.79±5.45 | 84.37±5.81 | 38.06±4.32 | 35.21±4.36 |
| **1e-5** | | | | | | | | |
| MAET_single_DE | 82.04±5.82 | 80.93±6.38 | 52.90±6.41 | 49.82±7.24 | 81.23±7.35 | 80.22±8.08 | 35.35±4.35 | 31.15±4.34 |
| MAET_single_PSD | 76.41±5.43 | 75.28±5.73 | 46.00±5.75 | 43.09±6.37 | 75.13±7.99 | 74.01±8.58 | 32.01±4.18 | 28.93±4.04 |
| MAET_single_SampEn | 58.55±5.49 | 57.52±5.59 | 30.57±2.90 | 29.12±2.93 | 65.50±6.66 | 64.96±6.82 | 27.80±3.65 | 26.32±3.24 |
| **1e-6** | | | | | | | | |
| MAET_single_DE | 62.25±4.25 | 59.69±4.63 | 29.77±2.59 | 26.01±3.00 | 51.61±2.41 | 49.11±2.88 | 21.39±1.60 | 18.82±1.74 |
| MAET_single_PSD | 59.33±4.27 | 57.20±4.59 | 27.75±2.79 | 24.79±3.27 | 50.90±1.84 | 49.00±2.05 | 20.58±1.23 | 18.62±1.32 |
| MAET_single_SampEn | 49.44±2.94 | 48.20±2.99 | 25.21±1.54 | 23.95±1.38 | 49.58±2.01 | 48.84±2.02 | 20.33±0.79 | 19.55±0.87 |

Table 11: MCA_overall

| | Dataset 1 (2-class) | | Dataset 1 (4-class) | | Dataset 2 (2-class) | | Dataset 2 (5-class) | |
|---|---|---|---|---|---|---|---|---|
| | ACC (%) | F1 (%) | ACC (%) | F1 (%) | ACC (%) | F1 (%) | ACC (%) | F1 (%) |
| **1e-3** | | | | | | | | |
| **MCA_PSD_DE** | **91.75±5.37** | **91.21±6.01** | **58.82±8.15** | **56.70±9.21** | **84.64±5.49** | **83.85±6.00** | **35.81±3.34** | **32.27±3.98** |
| MCA_SampEn_DE | 86.27±6.74 | 85.44±7.55 | 55.75±7.44 | 53.34±8.00 | 81.71±5.43 | 81.20±5.73 | 35.71±3.27 | 32.24±3.62 |
| MCA_SampEn_PSD | 83.46±6.31 | 82.84±6.70 | 51.74±6.58 | 49.59±7.56 | 77.58±6.23 | 76.89±6.61 | 33.12±3.35 | 30.21±3.29 |
| **1e-4** | | | | | | | | |
| MCA_PSD_DE | 88.82±5.86 | 88.11±6.40 | 56.58±8.79 | 54.44±9.67 | 82.70±6.09 | 81.90±6.66 | 35.62±3.50 | 32.05±3.92 |
| MCA_SampEn_DE | 77.92±9.77 | 75.72±11.64 | 51.67±9.20 | 48.06±10.00 | 79.29±5.68 | 78.53±6.22 | 35.44±3.52 | 31.54±4.08 |
| MCA_SampEn_PSD | 76.29±9.64 | 74.48±10.61 | 48.69±7.49 | 45.38±8.28 | 72.31±6.85 | 71.05±7.74 | 32.22±3.44 | 28.78±3.94 |
| **1e-5** | | | | | | | | |
| MCA_PSD_DE | 80.21±10.07 | 77.92±11.88 | 43.90±8.04 | 36.04±9.13 | 72.66±8.42 | 69.92±10.27 | 29.33±3.79 | 19.72±3.66 |
| MCA_SampEn_DE | 58.31±10.71 | 50.74±13.71 | 33.14±7.27 | 23.18±7.51 | 65.09±10.05 | 59.80±13.02 | 24.61±3.86 | 13.58±3.86 |
| MCA_SampEn_PSD | 61.48±9.90 | 55.43±12.11 | 30.05±5.07 | 21.17±5.96 | 59.93±8.85 | 53.26±11.91 | 23.17±3.65 | 12.31±4.23 |
| **1e-6** | | | | | | | | |
| MCA_PSD_DE | 55.11±3.42 | 46.90±3.47 | 27.32±1.96 | 17.35±2.10 | 59.19±5.76 | 51.74±6.57 | 23.18±2.43 | 14.62±1.93 |
| MCA_SampEn_DE | 51.56±3.47 | 43.60±3.47 | 26.21±2.18 | 15.38±2.25 | 57.32±5.20 | 49.98±5.83 | 22.44±1.34 | 12.90±1.30 |
| MCA_SampEn_PSD | 52.14±2.40 | 44.20±2.84 | 25.58±1.11 | 15.12±1.35 | 53.57±3.61 | 45.88±3.93 | 21.03±1.40 | 12.51±1.35 |

Table 12: SFT_overall

| | Dataset 1 (2-class) | | Dataset 1 (4-class) | | Dataset 2 (2-class) | | Dataset 2 (5-class) | |
|---|---|---|---|---|---|---|---|---|
| | ACC (%) | F1 (%) | ACC (%) | F1 (%) | ACC (%) | F1 (%) | ACC (%) | F1 (%) |
| **1e-3** | | | | | | | | |
| **SFT_16_DE** | **95.49±3.93** | **95.11±4.39** | **56.16±7.16** | **52.77±7.85** | **83.82±6.62** | **82.35±7.78** | **35.84±4.69** | **31.84±4.70** |
| SFT_16_PSD | 82.59±9.20 | 78.63±12.55 | 50.86±7.72 | 46.38±8.75 | 77.10±7.28 | 73.84±8.92 | 33.64±4.39 | 28.68±5.18 |
| SFT_16_SampEn | 80.07±11.35 | 75.26±14.74 | 43.23±7.47 | 39.58±8.64 | 74.24±6.16 | 71.97±7.00 | 31.52±3.75 | 28.73±3.78 |
| SFT_4_DE | 92.65±4.76 | 91.71±5.67 | 50.68±6.65 | 45.45±7.69 | 83.44±6.48 | 82.08±7.69 | 35.60±4.15 | 30.66±4.86 |
| SFT_4_PSD | 84.72±7.39 | 81.09±10.50 | 48.38±6.06 | 42.64±7.33 | 76.97±9.31 | 74.39±11.26 | 32.68±4.97 | 27.49±5.68 |
| SFT_4_SampEn | 71.80±9.44 | 63.02±12.86 | 41.40±6.99 | 35.88±8.33 | 76.80±6.70 | 75.35±7.42 | 31.56±3.51 | 27.60±4.07 |
| **1e-4** | | | | | | | | |
| SFT_16_DE | 61.42±2.67 | 45.50±4.45 | 36.08±4.65 | 25.03±5.74 | 69.51±9.69 | 63.91±12.79 | 32.27±4.53 | 25.02±5.36 |
| SFT_16_PSD | 58.65±3.22 | 41.28±5.91 | 34.80±4.06 | 23.63±5.64 | 61.15±8.18 | 52.37±11.25 | 27.46±4.44 | 18.50±5.49 |
| SFT_16_SampEn | 55.70±0.53 | 35.77±0.22 | 27.90±1.53 | 12.87±1.84 | 53.72±5.47 | 40.20±7.94 | 23.13±3.52 | 12.58±4.52 |
| SFT_4_DE | 54.51±1.27 | 36.70±2.00 | 28.48±1.27 | 14.07±1.63 | 58.99±6.02 | 47.84±8.24 | 26.06±3.73 | 14.63±4.22 |
| SFT_4_PSD | 54.70±1.04 | 37.16±1.80 | 27.98±1.47 | 13.64±2.05 | 54.90±5.42 | 42.60±7.55 | 23.72±3.08 | 12.36±3.70 |
| SFT_4_SampEn | 53.45±0.32 | 34.82±0.22 | 26.22±0.43 | 11.09±0.57 | 51.73±2.42 | 37.10±3.96 | 21.10±1.22 | 9.12±1.41 |
| **1e-5** | | | | | | | | |
| SFT_16_DE | 48.88±0.20 | 32.80±0.24 | 27.20±0.94 | 12.74±1.08 | 50.70±0.94 | 34.77±1.65 | 20.60±0.56 | 7.77±0.63 |
| SFT_16_PSD | 48.87±0.21 | 32.83±0.22 | 26.77±0.94 | 12.37±0.96 | 50.86±1.71 | 35.29±2.37 | 20.15±0.43 | 7.41±0.53 |
| SFT_16_SampEn | 49.04±0.52 | 32.96±0.55 | 25.77±0.68 | 10.93±0.52 | 50.17±0.34 | 33.81±0.66 | 20.36±0.27 | 7.42±0.23 |
| SFT_4_DE | 52.72±0.29 | 34.54±0.29 | 24.91±0.07 | 9.96±0.03 | 50.28±0.95 | 34.26±1.44 | 20.44±0.52 | 7.43±0.49 |
| SFT_4_PSD | 52.87±0.47 | 34.90±0.72 | 24.90±0.11 | 9.98±0.07 | 49.96±0.92 | 33.85±1.14 | 20.11±0.48 | 7.20±0.42 |
| SFT_4_SampEn | 52.67±0.26 | 34.42±0.10 | 24.92±0.11 | 9.99±0.15 | 50.08±0.20 | 33.83±0.41 | 20.06±0.12 | 7.04±0.21 |
| **1e-6** | | | | | | | | |
| SFT_16_DE | 48.74±0.39 | 32.74±0.29 | 24.84±0.57 | 10.35±0.41 | 50.27±0.64 | 34.06±0.93 | 20.05±0.30 | 7.41±0.34 |
| SFT_16_PSD | 48.64±0.51 | 32.71±0.34 | 24.97±0.54 | 10.53±0.47 | 50.09±0.44 | 33.82±0.71 | 20.03±0.40 | 7.25±0.28 |
| SFT_16_SampEn | 48.86±0.17 | 32.73±0.09 | 24.81±0.28 | 10.35±0.18 | 50.01±0.03 | 33.36±0.06 | 20.24±0.23 | 7.47±0.24 |
| SFT_4_DE | 52.70±0.26 | 34.50±0.23 | 24.91±0.07 | 9.95±0.03 | 49.97±0.25 | 33.44±0.26 | 20.01±0.26 | 6.90±0.21 |
| SFT_4_PSD | 52.74±0.42 | 34.65±0.53 | 24.89±0.10 | 9.95±0.03 | 50.01±0.08 | 33.39±0.08 | 20.01±0.21 | 6.85±0.21 |
| SFT_4_SampEn | 52.67±0.27 | 34.44±0.14 | 24.91±0.07 | 9.96±0.04 | 50.02±0.05 | 33.46±0.15 | 19.98±0.09 | 6.81±0.11 |

Table 13: GACET_overall

| | Dataset 1 (2-class) | | Dataset 1 (4-class) | | Dataset 2 (2-class) | | Dataset 2 (5-class) | |
|---|---|---|---|---|---|---|---|---|
| | ACC (%) | F1 (%) | ACC (%) | F1 (%) | ACC (%) | F1 (%) | ACC (%) | F1 (%) |
| **1e-3** | | | | | | | | |
| GACET_PSD_DE | 95.98±3.92 | 95.70±4.49 | 61.14±8.15 | 57.18±9.11 | 87.83±5.49 | 87.02±6.52 | 35.17±2.67 | 25.40±3.82 |
| GACET_SampEn_DE | 95.73±3.95 | 95.47±4.54 | 61.11±8.19 | 57.18±8.83 | 89.90±3.67 | 89.64±3.87 | 36.54±3.23 | 26.80±4.67 |
| GACET_SampEn_PSD | 95.90±4.08 | 95.67±4.54 | 60.31±6.78 | 57.42±7.39 | 87.61±4.37 | 87.23±4.67 | 35.72±3.09 | 26.59±4.56 |
| **1e-4** | | | | | | | | |
| GACET_PSD_DE | 97.33±3.26 | 97.18±3.71 | 65.40±9.37 | 61.85±10.67 | 90.45±5.12 | 89.88±6.07 | 40.78±4.32 | 35.73±5.04 |
| **GACET_SampEn_DE** | **97.34±2.96** | **97.20±3.31** | **67.66±9.03** | **64.46±10.19** | **92.75±3.76** | **92.52±4.10** | **42.77±4.29** | **38.01±5.22** |
| GACET_SampEn_PSD | 97.69±2.28 | 97.62±2.37 | 66.29±8.60 | 63.57±9.67 | 92.04±3.57 | 91.88±3.74 | 41.73±3.76 | 37.61±4.37 |
| **1e-5** | | | | | | | | |
| GACET_PSD_DE | 94.86±4.13 | 94.52±4.66 | 61.00±8.93 | 57.81±9.83 | 89.40±4.79 | 88.94±5.29 | 38.00±3.38 | 32.46±3.68 |
| GACET_SampEn_DE | 95.07±3.98 | 94.84±4.33 | 60.98±8.31 | 58.20±9.11 | 90.56±3.45 | 90.25±3.74 | 38.87±2.98 | 33.20±3.41 |
| GACET_SampEn_PSD | 94.63±3.66 | 94.40±3.90 | 59.19±7.77 | 56.79±8.55 | 88.78±3.99 | 88.39±4.35 | 37.46±2.56 | 32.48±2.62 |
| **1e-6** | | | | | | | | |
| GACET_PSD_DE | 65.90±7.31 | 61.03±8.45 | 29.51±2.93 | 23.17±3.25 | 75.33±9.08 | 73.33±10.16 | 24.22±2.67 | 19.99±3.17 |
| GACET_SampEn_DE | 65.37±6.00 | 60.55±7.23 | 30.33±3.05 | 24.57±3.23 | 74.24±8.66 | 72.23±9.88 | 23.85±2.68 | 20.05±3.00 |
| GACET_SampEn_PSD | 61.70±4.21 | 56.64±5.11 | 29.86±2.04 | 24.24±2.24 | 68.13±6.92 | 66.03±7.76 | 22.89±1.90 | 19.33±2.05 |

# H GACET PERFORMANCE DETAILS

## H.1 PERFORMANCE DETAILS AT DIFFERENT LEARNING RATES

### H.1.1 1E-3

Table 14: Dataset 1 (2class)

| Subject | Cross Acc (%) | Cross F1 (%) | In-time Acc (%) | In-time F1 (%) |
|---|---|---|---|---|
| Subject_1 | 100.00 | 100.00 | 100.00 | 100.00 |
| Subject_2 | 92.10 | 91.84 | 100.00 | 100.00 |
| Subject_3 | 99.80 | 99.80 | 100.00 | 100.00 |
| Subject_4 | 99.19 | 99.18 | 100.00 | 100.00 |
| Subject_5 | 98.54 | 98.54 | 100.00 | 100.00 |
| Subject_6 | 90.74 | 90.42 | 97.78 | 97.74 |
| Subject_7 | 96.46 | 96.36 | 100.00 | 100.00 |
| Subject_8 | 96.46 | 96.27 | 100.00 | 100.00 |
| Subject_9 | 98.96 | 98.94 | 96.67 | 96.52 |
| Subject_10 | 98.15 | 98.11 | 98.02 | 98.01 |
| Subject_11 | 95.96 | 95.94 | 100.00 | 100.00 |
| Subject_12 | 97.95 | 97.93 | 99.44 | 99.44 |
| Subject_13 | 95.83 | 95.63 | 99.44 | 99.33 |
| Subject_14 | 95.00 | 94.81 | 100.00 | 100.00 |
| Subject_15 | 98.54 | 98.50 | 100.00 | 100.00 |
| Subject_16 | 95.83 | 95.60 | 100.00 | 100.00 |
| Subject_17 | 94.70 | 94.61 | 100.00 | 100.00 |
| Subject_18 | 100.00 | 100.00 | 100.00 | 100.00 |
| Subject_19 | 91.88 | 91.31 | 97.86 | 97.74 |
| Subject_20 | 94.66 | 94.56 | 96.11 | 95.96 |
| Subject_21 | 90.83 | 90.59 | 98.33 | 98.32 |
| Subject_22 | 96.75 | 96.65 | 96.67 | 96.52 |
| Subject_23 | 90.06 | 89.07 | 97.78 | 97.71 |
| Subject_24 | 98.96 | 98.95 | 100.00 | 100.00 |
| Subject_25 | 96.47 | 96.41 | 99.44 | 99.44 |
| Subject_26 | 97.10 | 97.01 | 98.89 | 98.88 |
| Subject_27 | 94.70 | 94.61 | 100.00 | 100.00 |
| Subject_28 | 81.25 | 77.50 | 100.00 | 100.00 |
| Subject_29 | 99.38 | 99.37 | 99.44 | 99.44 |
| Summary | 95.73±3.95 | 95.47±4.54 | 99.17±1.19 | 99.14±1.23 |

Table 15: Dataset 1 (4class)

| Subject | Cross Acc (%) | Cross F1 (%) | In-time Acc (%) | In-time F1 (%) |
|---|---|---|---|---|
| Subject_1 | 65.05 | 61.59 | 98.02 | 98.17 |
| Subject_2 | 62.07 | 56.37 | 92.05 | 92.17 |
| Subject_3 | 66.14 | 63.09 | 97.36 | 97.44 |
| Subject_4 | 72.75 | 68.85 | 95.13 | 95.11 |
| Subject_5 | 61.57 | 55.71 | 95.88 | 95.27 |
| Subject_6 | 54.91 | 48.63 | 93.06 | 92.50 |
| Subject_7 | 68.24 | 63.45 | 98.75 | 98.75 |
| Subject_8 | 58.82 | 56.40 | 96.54 | 96.63 |
| Subject_9 | 57.06 | 55.31 | 96.72 | 96.29 |
| Subject_10 | 52.25 | 49.90 | 97.09 | 97.04 |
| Subject_11 | 70.82 | 67.60 | 96.81 | 96.74 |
| Subject_12 | 65.13 | 61.63 | 98.04 | 97.97 |

*Continued on next page*

*Continued from previous page*

| Subject | Cross Acc (%) | Cross F1 (%) | In-time Acc (%) | In-time F1 (%) |
|---|---|---|---|---|
| Subject_13 | 50.73 | 44.83 | 86.65 | 85.29 |
| Subject_14 | 54.51 | 50.63 | 95.27 | 95.61 |
| Subject_15 | 63.43 | 57.95 | 94.63 | 94.33 |
| Subject_16 | 63.92 | 59.90 | 98.57 | 98.64 |
| Subject_17 | 63.67 | 59.15 | 98.26 | 98.03 |
| Subject_18 | 62.30 | 61.13 | 98.77 | 98.88 |
| Subject_19 | 46.86 | 44.20 | 94.87 | 94.47 |
| Subject_20 | 65.07 | 60.76 | 88.21 | 87.40 |
| Subject_21 | 74.12 | 72.00 | 96.94 | 96.67 |
| Subject_22 | 52.00 | 46.70 | 95.57 | 95.58 |
| Subject_23 | 55.38 | 48.73 | 97.05 | 96.93 |
| Subject_24 | 65.69 | 63.12 | 98.79 | 98.77 |
| Subject_25 | 75.54 | 74.11 | 98.53 | 98.37 |
| Subject_26 | 53.51 | 52.70 | 93.44 | 93.42 |
| Subject_27 | 63.67 | 59.15 | 98.26 | 98.03 |
| Subject_28 | 39.71 | 32.75 | 94.63 | 94.60 |
| Subject_29 | 67.41 | 62.04 | 85.68 | 83.31 |
| Summary | 61.11±8.19 | 57.18±8.83 | 95.50±3.45 | 95.26±3.87 |

Table 16: Dataset 2 (2class)

| Subject | Cross Acc (%) | Cross F1 (%) | In-time Acc (%) | In-time F1 (%) |
|---|---|---|---|---|
| Subject_1 | 91.90 | 91.70 | 99.60 | 99.59 |
| Subject_2 | 93.81 | 93.76 | 99.00 | 98.99 |
| Subject_3 | 83.02 | 82.17 | 97.05 | 96.98 |
| Subject_4 | 89.29 | 89.10 | 99.61 | 99.60 |
| Subject_5 | 89.52 | 89.43 | 97.84 | 97.83 |
| Subject_6 | 87.86 | 87.63 | 97.22 | 97.21 |
| Subject_7 | 85.00 | 84.75 | 95.04 | 94.90 |
| Subject_8 | 96.43 | 96.42 | 100.00 | 100.00 |
| Subject_9 | 89.68 | 89.59 | 99.40 | 99.40 |
| Subject_10 | 87.22 | 86.49 | 98.01 | 97.99 |
| Subject_11 | 95.71 | 95.69 | 100.00 | 100.00 |
| Subject_12 | 90.40 | 90.29 | 98.59 | 98.59 |
| Subject_13 | 91.43 | 91.23 | 99.22 | 99.22 |
| Subject_14 | 87.30 | 86.67 | 98.81 | 98.80 |
| Summary | 89.90±3.67 | 89.64±3.87 | 98.53±1.34 | 98.51±1.37 |

Table 17: Dataset 2 (5class)

| Subject | Cross Acc (%) | Cross F1 (%) | In-time Acc (%) | In-time F1 (%) |
|---|---|---|---|---|
| Subject_1 | 36.63 | 27.39 | 45.16 | 36.57 |
| Subject_2 | 40.32 | 33.04 | 53.49 | 46.20 |
| Subject_3 | 32.06 | 21.02 | 40.40 | 30.93 |
| Subject_4 | 39.33 | 30.96 | 50.48 | 44.00 |
| Subject_5 | 38.51 | 30.69 | 53.33 | 48.34 |
| Subject_6 | 33.08 | 22.40 | 41.35 | 29.27 |
| Subject_7 | 32.98 | 21.38 | 39.92 | 28.77 |
| Subject_8 | 39.14 | 26.75 | 43.49 | 32.77 |
| Subject_9 | 32.86 | 21.70 | 44.52 | 34.27 |

*Continued on next page*

*Continued from previous page*

| Subject | Cross Acc (%) | Cross F1 (%) | In-time Acc (%) | In-time F1 (%) |
|---|---|---|---|---|
| Subject_10 | 36.98 | 27.21 | 44.52 | 35.48 |
| Subject_11 | 42.51 | 35.26 | 52.86 | 44.59 |
| Subject_12 | 32.98 | 21.64 | 43.41 | 33.69 |
| Subject_13 | 35.30 | 24.32 | 45.08 | 36.01 |
| Subject_14 | 38.83 | 31.40 | 53.65 | 45.36 |
| Summary | 36.54±3.23 | 26.80±4.67 | 46.55±4.93 | 37.59±6.49 |

### H.1.2 1E-4

Table 18: Dataset 1 (2class)

| Subject | Cross Acc (%) | Cross F1 (%) | In-time Acc (%) | In-time F1 (%) |
|---|---|---|---|---|
| Subject_1 | 100.00 | 100.00 | 100.00 | 100.00 |
| Subject_2 | 98.78 | 98.76 | 97.78 | 97.66 |
| Subject_3 | 100.00 | 100.00 | 100.00 | 100.00 |
| Subject_4 | 100.00 | 100.00 | 100.00 | 100.00 |
| Subject_5 | 100.00 | 100.00 | 100.00 | 100.00 |
| Subject_6 | 92.32 | 92.04 | 98.33 | 98.30 |
| Subject_7 | 98.75 | 98.75 | 100.00 | 100.00 |
| Subject_8 | 98.75 | 98.73 | 99.44 | 99.44 |
| Subject_9 | 98.33 | 98.28 | 96.11 | 95.94 |
| Subject_10 | 98.78 | 98.76 | 99.44 | 99.44 |
| Subject_11 | 97.98 | 97.96 | 98.89 | 98.88 |
| Subject_12 | 97.58 | 97.50 | 97.78 | 97.74 |
| Subject_13 | 98.14 | 98.10 | 100.00 | 100.00 |
| Subject_14 | 92.92 | 92.80 | 99.52 | 99.52 |
| Subject_15 | 98.96 | 98.93 | 100.00 | 100.00 |
| Subject_16 | 94.17 | 93.95 | 98.89 | 98.86 |
| Subject_17 | 97.97 | 97.95 | 100.00 | 100.00 |
| Subject_18 | 100.00 | 100.00 | 100.00 | 100.00 |
| Subject_19 | 95.62 | 95.46 | 96.67 | 96.44 |
| Subject_20 | 95.52 | 95.39 | 93.89 | 93.39 |
| Subject_21 | 94.79 | 94.71 | 97.78 | 97.66 |
| Subject_22 | 96.14 | 96.06 | 93.89 | 93.08 |
| Subject_23 | 95.48 | 95.40 | 98.33 | 98.30 |
| Subject_24 | 100.00 | 100.00 | 100.00 | 100.00 |
| Subject_25 | 98.99 | 98.96 | 99.44 | 99.44 |
| Subject_26 | 98.34 | 98.29 | 95.56 | 95.16 |
| Subject_27 | 97.97 | 97.95 | 100.00 | 100.00 |
| Subject_28 | 86.67 | 84.18 | 100.00 | 100.00 |
| Subject_29 | 100.00 | 100.00 | 100.00 | 100.00 |
| Summary | 97.34±2.96 | 97.20±3.31 | 98.68±1.80 | 98.59±1.97 |

Table 19: Dataset 1 (4class)

| Subject | Cross Acc (%) | Cross F1 (%) | In-time Acc (%) | In-time F1 (%) |
|---|---|---|---|---|
| Subject_1 | 68.94 | 66.70 | 98.52 | 98.61 |
| Subject_2 | 71.27 | 68.56 | 96.04 | 95.44 |
| Subject_3 | 75.00 | 73.43 | 96.17 | 96.24 |
| Subject_4 | 77.00 | 73.15 | 96.59 | 96.70 |

*Continued on next page*

*Continued from previous page*

| Subject | Cross Acc (%) | Cross F1 (%) | In-time Acc (%) | In-time F1 (%) |
|---|---|---|---|---|
| Subject_5 | 73.14 | 66.98 | 98.79 | 98.86 |
| Subject_6 | 61.60 | 56.15 | 93.00 | 92.92 |
| Subject_7 | 77.65 | 74.11 | 97.44 | 96.79 |
| Subject_8 | 66.27 | 62.01 | 98.00 | 98.12 |
| Subject_9 | 67.06 | 63.89 | 96.72 | 96.45 |
| Subject_10 | 58.79 | 56.79 | 95.64 | 95.65 |
| Subject_11 | 74.20 | 70.78 | 97.77 | 97.73 |
| Subject_12 | 75.01 | 73.37 | 96.08 | 95.99 |
| Subject_13 | 49.25 | 43.23 | 95.59 | 95.73 |
| Subject_14 | 56.47 | 53.82 | 98.99 | 99.05 |
| Subject_15 | 70.98 | 67.84 | 97.51 | 97.43 |
| Subject_16 | 73.63 | 72.60 | 98.79 | 98.78 |
| Subject_17 | 74.26 | 72.28 | 97.78 | 97.56 |
| Subject_18 | 64.81 | 64.00 | 99.29 | 99.37 |
| Subject_19 | 51.47 | 46.74 | 93.96 | 93.84 |
| Subject_20 | 67.21 | 63.17 | 91.10 | 90.56 |
| Subject_21 | 83.04 | 81.91 | 98.74 | 98.49 |
| Subject_22 | 55.42 | 51.40 | 95.75 | 95.50 |
| Subject_23 | 64.13 | 59.14 | 97.49 | 97.26 |
| Subject_24 | 70.49 | 67.01 | 98.55 | 98.57 |
| Subject_25 | 80.90 | 79.63 | 99.27 | 99.20 |
| Subject_26 | 62.03 | 60.70 | 90.53 | 90.52 |
| Subject_27 | 74.26 | 72.28 | 97.78 | 97.56 |
| Subject_28 | 48.24 | 39.39 | 98.26 | 98.15 |
| Subject_29 | 69.47 | 68.29 | 98.75 | 98.80 |
| Summary | 67.66±9.03 | 64.46±10.19 | 96.86±2.24 | 96.75±2.31 |

Table 20: Dataset 2 (2class)

| Subject | Cross Acc (%) | Cross F1 (%) | In-time Acc (%) | In-time F1 (%) |
|---|---|---|---|---|
| Subject_1 | 92.46 | 92.26 | 98.81 | 98.80 |
| Subject_2 | 96.35 | 96.34 | 98.59 | 98.58 |
| Subject_3 | 82.30 | 81.03 | 97.25 | 97.23 |
| Subject_4 | 90.32 | 90.12 | 98.42 | 98.40 |
| Subject_5 | 92.22 | 92.10 | 98.24 | 98.22 |
| Subject_6 | 95.79 | 95.78 | 99.41 | 99.41 |
| Subject_7 | 93.10 | 92.94 | 98.00 | 97.99 |
| Subject_8 | 98.57 | 98.57 | 100.00 | 100.00 |
| Subject_9 | 93.89 | 93.85 | 99.80 | 99.80 |
| Subject_10 | 93.10 | 93.07 | 98.82 | 98.82 |
| Subject_11 | 95.24 | 95.21 | 99.22 | 99.22 |
| Subject_12 | 91.90 | 91.84 | 98.39 | 98.39 |
| Subject_13 | 94.44 | 94.39 | 98.24 | 98.23 |
| Subject_14 | 88.81 | 87.81 | 99.40 | 99.40 |
| Summary | 92.75±3.76 | 92.52±4.10 | 98.76±0.72 | 98.75±0.73 |

Table 21: Dataset 2 (5class)

| Subject | Cross Acc (%) | Cross F1 (%) | In-time Acc (%) | In-time F1 (%) |
|---|---|---|---|---|
| Subject_1 | 44.41 | 39.99 | 61.67 | 59.66 |

*Continued on next page*

| Subject | Cross Acc (%) | Cross F1 (%) | In-time Acc (%) | In-time F1 (%) |
|---|---|---|---|---|
| *Continued from previous page* | | | | |
| Subject_2 | 44.16 | 41.07 | 62.38 | 60.39 |
| Subject_3 | 36.44 | 30.06 | 48.97 | 46.43 |
| Subject_4 | 46.95 | 43.50 | 59.05 | 56.63 |
| Subject_5 | 40.19 | 35.19 | 60.08 | 57.73 |
| Subject_6 | 40.19 | 36.15 | 58.89 | 56.28 |
| Subject_7 | 36.44 | 31.55 | 52.94 | 50.03 |
| Subject_8 | 49.59 | 46.44 | 57.06 | 55.13 |
| Subject_9 | 42.57 | 37.77 | 59.76 | 57.31 |
| Subject_10 | 42.89 | 37.33 | 54.92 | 53.11 |
| Subject_11 | 47.68 | 43.42 | 65.08 | 63.94 |
| Subject_12 | 40.89 | 35.20 | 58.57 | 57.07 |
| Subject_13 | 37.40 | 29.80 | 55.00 | 50.74 |
| Subject_14 | 49.02 | 44.75 | 62.78 | 60.43 |
| Summary | 42.77±4.29 | 38.01±5.22 | 58.37±4.14 | 56.06±4.52 |

### H.1.3    1E-5

Table 22: Dataset 1 (2class)

| Subject | Cross Acc (%) | Cross F1 (%) | In-time Acc (%) | In-time F1 (%) |
|---|---|---|---|---|
| Subject_1 | 98.99 | 98.98 | 100.00 | 100.00 |
| Subject_2 | 93.54 | 93.26 | 96.75 | 96.60 |
| Subject_3 | 96.35 | 96.29 | 99.44 | 99.44 |
| Subject_4 | 98.99 | 98.97 | 99.44 | 99.44 |
| Subject_5 | 99.17 | 99.15 | 100.00 | 100.00 |
| Subject_6 | 91.12 | 90.97 | 95.56 | 95.33 |
| Subject_7 | 98.75 | 98.73 | 97.22 | 97.08 |
| Subject_8 | 95.21 | 95.10 | 100.00 | 100.00 |
| Subject_9 | 96.46 | 96.32 | 96.67 | 96.38 |
| Subject_10 | 98.36 | 98.32 | 99.44 | 99.44 |
| Subject_11 | 94.14 | 94.08 | 98.89 | 98.88 |
| Subject_12 | 95.93 | 95.78 | 95.56 | 95.38 |
| Subject_13 | 86.40 | 85.19 | 100.00 | 100.00 |
| Subject_14 | 83.96 | 82.27 | 97.86 | 97.74 |
| Subject_15 | 98.96 | 98.93 | 100.00 | 100.00 |
| Subject_16 | 93.12 | 92.84 | 98.97 | 98.94 |
| Subject_17 | 95.92 | 95.80 | 98.33 | 98.30 |
| Subject_18 | 99.17 | 99.15 | 100.00 | 100.00 |
| Subject_19 | 93.96 | 93.82 | 96.11 | 95.77 |
| Subject_20 | 94.48 | 94.30 | 92.78 | 92.02 |
| Subject_21 | 92.92 | 92.77 | 97.78 | 97.66 |
| Subject_22 | 93.66 | 93.47 | 94.44 | 93.72 |
| Subject_23 | 90.56 | 90.18 | 96.11 | 96.01 |
| Subject_24 | 99.58 | 99.58 | 100.00 | 100.00 |
| Subject_25 | 97.56 | 97.45 | 98.89 | 98.88 |
| Subject_26 | 97.30 | 97.22 | 96.67 | 96.08 |
| Subject_27 | 95.92 | 95.80 | 98.33 | 98.30 |
| Subject_28 | 87.71 | 86.64 | 97.78 | 97.66 |
| Subject_29 | 98.96 | 98.94 | 98.89 | 98.86 |
| Summary | 95.07±3.98 | 94.84±4.33 | 98.00±1.88 | 97.86±2.06 |

Table 23: Dataset 1 (4class)

| Subject | Cross Acc (%) | Cross F1 (%) | In-time Acc (%) | In-time F1 (%) |
|---|---|---|---|---|
| Subject_1 | 64.58 | 62.00 | 87.66 | 87.22 |
| Subject_2 | 64.20 | 60.76 | 86.19 | 83.68 |
| Subject_3 | 67.50 | 65.44 | 94.65 | 94.79 |
| Subject_4 | 69.18 | 66.13 | 88.86 | 89.00 |
| Subject_5 | 61.86 | 56.51 | 92.62 | 92.54 |
| Subject_6 | 58.32 | 54.97 | 87.18 | 86.08 |
| Subject_7 | 68.33 | 65.88 | 91.87 | 90.68 |
| Subject_8 | 62.94 | 59.41 | 91.12 | 91.22 |
| Subject_9 | 62.06 | 60.28 | 91.30 | 90.90 |
| Subject_10 | 55.46 | 53.47 | 91.23 | 90.78 |
| Subject_11 | 70.05 | 67.54 | 93.74 | 93.60 |
| Subject_12 | 64.50 | 63.57 | 86.74 | 86.77 |
| Subject_13 | 47.18 | 42.50 | 86.19 | 86.36 |
| Subject_14 | 52.45 | 49.69 | 89.71 | 89.87 |
| Subject_15 | 63.73 | 59.78 | 86.37 | 86.03 |
| Subject_16 | 65.29 | 64.63 | 95.84 | 95.62 |
| Subject_17 | 67.28 | 65.79 | 90.20 | 89.72 |
| Subject_18 | 58.90 | 55.52 | 92.14 | 92.29 |
| Subject_19 | 49.02 | 45.57 | 88.59 | 87.58 |
| Subject_20 | 65.76 | 62.45 | 81.28 | 79.73 |
| Subject_21 | 74.22 | 71.69 | 89.32 | 87.15 |
| Subject_22 | 46.43 | 44.92 | 88.94 | 89.03 |
| Subject_23 | 51.12 | 46.47 | 90.92 | 91.00 |
| Subject_24 | 58.73 | 55.55 | 87.11 | 85.76 |
| Subject_25 | 73.79 | 72.48 | 95.57 | 95.69 |
| Subject_26 | 55.60 | 54.16 | 81.21 | 81.11 |
| Subject_27 | 67.28 | 65.79 | 90.20 | 89.72 |
| Subject_28 | 39.71 | 32.61 | 90.48 | 90.38 |
| Subject_29 | 63.03 | 62.17 | 89.95 | 90.11 |
| Summary | 60.98±8.31 | 58.20±9.11 | 89.56±3.47 | 89.12±3.78 |

Table 24: Dataset 2 (2class)

| Subject | Cross Acc (%) | Cross F1 (%) | In-time Acc (%) | In-time F1 (%) |
|---|---|---|---|---|
| Subject_1 | 93.33 | 93.29 | 96.41 | 96.38 |
| Subject_2 | 94.05 | 93.99 | 97.01 | 96.99 |
| Subject_3 | 83.89 | 83.22 | 93.31 | 93.11 |
| Subject_4 | 91.27 | 91.16 | 97.05 | 97.00 |
| Subject_5 | 89.60 | 89.47 | 94.51 | 94.32 |
| Subject_6 | 89.52 | 89.41 | 96.46 | 96.44 |
| Subject_7 | 87.14 | 86.37 | 92.28 | 91.98 |
| Subject_8 | 96.90 | 96.89 | 99.22 | 99.22 |
| Subject_9 | 92.46 | 92.36 | 99.02 | 99.02 |
| Subject_10 | 87.22 | 86.11 | 97.45 | 97.44 |
| Subject_11 | 94.05 | 94.00 | 97.65 | 97.64 |
| Subject_12 | 90.95 | 90.87 | 98.62 | 98.61 |
| Subject_13 | 91.35 | 91.19 | 97.45 | 97.45 |
| Subject_14 | 86.11 | 85.23 | 94.88 | 94.74 |
| Summary | 90.56±3.45 | 90.25±3.74 | 96.52±2.01 | 96.45±2.10 |

Table 25: Dataset 2 (5class)

| Subject | Cross Acc (%) | Cross F1 (%) | In-time Acc (%) | In-time F1 (%) |
|---|---|---|---|---|
| Subject_1 | 40.51 | 34.94 | 51.19 | 47.21 |
| Subject_2 | 40.92 | 35.61 | 50.24 | 45.69 |
| Subject_3 | 34.41 | 28.70 | 42.70 | 38.59 |
| Subject_4 | 42.83 | 38.13 | 47.30 | 41.82 |
| Subject_5 | 36.89 | 30.62 | 48.33 | 43.17 |
| Subject_6 | 35.43 | 29.88 | 41.75 | 37.01 |
| Subject_7 | 34.25 | 29.06 | 39.52 | 34.87 |
| Subject_8 | 42.98 | 39.03 | 47.46 | 43.97 |
| Subject_9 | 39.71 | 32.79 | 49.68 | 45.67 |
| Subject_10 | 37.27 | 30.94 | 45.08 | 42.03 |
| Subject_11 | 43.43 | 38.42 | 51.51 | 47.81 |
| Subject_12 | 37.40 | 31.72 | 44.92 | 40.82 |
| Subject_13 | 38.10 | 30.85 | 48.33 | 43.40 |
| Subject_14 | 40.00 | 34.12 | 50.71 | 46.63 |
| Summary | 38.87±2.98 | 33.20±3.41 | 47.05±3.62 | 42.76±3.75 |

### H.1.4   1E-6

Table 26: Dataset 1 (2class)

| Subject | Cross Acc (%) | Cross F1 (%) | In-time Acc (%) | In-time F1 (%) |
|---|---|---|---|---|
| Subject_1 | 70.23 | 66.05 | 80.87 | 76.45 |
| Subject_2 | 58.38 | 52.20 | 65.63 | 60.36 |
| Subject_3 | 61.48 | 54.70 | 72.22 | 67.53 |
| Subject_4 | 69.29 | 65.16 | 78.81 | 76.43 |
| Subject_5 | 58.75 | 53.45 | 74.44 | 70.85 |
| Subject_6 | 62.83 | 58.37 | 71.19 | 68.57 |
| Subject_7 | 67.08 | 64.26 | 73.89 | 70.36 |
| Subject_8 | 63.33 | 58.50 | 73.33 | 69.76 |
| Subject_9 | 74.38 | 72.09 | 78.02 | 74.23 |
| Subject_10 | 64.13 | 57.09 | 70.63 | 65.47 |
| Subject_11 | 59.19 | 55.42 | 74.13 | 72.17 |
| Subject_12 | 64.76 | 57.93 | 78.97 | 75.00 |
| Subject_13 | 48.77 | 39.47 | 63.65 | 57.18 |
| Subject_14 | 64.17 | 58.08 | 75.40 | 71.16 |
| Subject_15 | 77.29 | 74.98 | 85.16 | 81.37 |
| Subject_16 | 60.00 | 54.50 | 69.52 | 65.08 |
| Subject_17 | 66.00 | 62.10 | 78.17 | 74.94 |
| Subject_18 | 72.27 | 69.13 | 84.76 | 81.78 |
| Subject_19 | 72.50 | 68.66 | 73.81 | 69.48 |
| Subject_20 | 61.86 | 55.72 | 68.17 | 62.85 |
| Subject_21 | 56.46 | 50.80 | 63.49 | 58.28 |
| Subject_22 | 67.45 | 61.98 | 74.68 | 69.87 |
| Subject_23 | 67.03 | 62.82 | 78.81 | 74.47 |
| Subject_24 | 63.12 | 58.65 | 71.03 | 65.67 |
| Subject_25 | 71.72 | 67.57 | 81.11 | 79.51 |
| Subject_26 | 69.84 | 65.37 | 73.73 | 69.28 |
| Subject_27 | 66.00 | 62.10 | 78.17 | 74.94 |
| Subject_28 | 65.00 | 60.11 | 73.81 | 70.70 |
| Subject_29 | 72.41 | 68.80 | 79.21 | 75.97 |
| Summary | 65.37±6.00 | 60.55±7.23 | 74.65±5.42 | 70.68±6.16 |

Table 27: Dataset 1 (4class)

| Subject | Cross Acc (%) | Cross F1 (%) | In-time Acc (%) | In-time F1 (%) |
|---|---|---|---|---|
| Subject_1 | 30.55 | 24.65 | 39.65 | 34.87 |
| Subject_2 | 27.57 | 21.00 | 38.11 | 32.04 |
| Subject_3 | 33.99 | 27.28 | 39.38 | 33.10 |
| Subject_4 | 35.17 | 30.23 | 39.07 | 34.85 |
| Subject_5 | 23.24 | 17.52 | 44.89 | 39.99 |
| Subject_6 | 32.43 | 25.13 | 43.02 | 38.47 |
| Subject_7 | 35.69 | 30.21 | 41.30 | 34.60 |
| Subject_8 | 30.39 | 23.98 | 40.22 | 36.38 |
| Subject_9 | 29.31 | 24.31 | 44.25 | 37.97 |
| Subject_10 | 32.40 | 26.34 | 43.24 | 36.68 |
| Subject_11 | 31.11 | 25.10 | 41.70 | 35.77 |
| Subject_12 | 30.24 | 26.42 | 36.48 | 31.74 |
| Subject_13 | 28.67 | 20.89 | 35.88 | 28.96 |
| Subject_14 | 27.75 | 21.71 | 37.05 | 32.10 |
| Subject_15 | 31.96 | 27.43 | 39.45 | 35.66 |
| Subject_16 | 30.59 | 24.68 | 34.40 | 28.54 |
| Subject_17 | 32.82 | 26.65 | 39.01 | 33.52 |
| Subject_18 | 28.46 | 22.48 | 38.17 | 33.16 |
| Subject_19 | 30.00 | 24.61 | 41.79 | 36.00 |
| Subject_20 | 29.86 | 23.55 | 37.01 | 32.42 |
| Subject_21 | 34.61 | 29.78 | 37.88 | 31.61 |
| Subject_22 | 31.41 | 26.14 | 37.86 | 33.11 |
| Subject_23 | 28.40 | 22.95 | 43.64 | 38.44 |
| Subject_24 | 27.55 | 22.28 | 36.10 | 30.51 |
| Subject_25 | 32.46 | 27.67 | 36.83 | 32.56 |
| Subject_26 | 28.57 | 22.32 | 39.25 | 32.81 |
| Subject_27 | 32.82 | 26.65 | 39.01 | 33.52 |
| Subject_28 | 22.06 | 16.48 | 31.32 | 25.94 |
| Subject_29 | 29.56 | 24.19 | 36.54 | 31.39 |
| Summary | 30.33±3.05 | 24.57±3.23 | 39.05±3.04 | 33.68±3.10 |

Table 28: Dataset 2 (2class)

| Subject | Cross Acc (%) | Cross F1 (%) | In-time Acc (%) | In-time F1 (%) |
|---|---|---|---|---|
| Subject_1 | 73.49 | 71.94 | 79.11 | 76.85 |
| Subject_2 | 84.37 | 83.95 | 91.79 | 91.70 |
| Subject_3 | 64.13 | 60.96 | 72.95 | 71.52 |
| Subject_4 | 79.92 | 79.27 | 87.78 | 86.88 |
| Subject_5 | 62.94 | 59.24 | 69.25 | 65.97 |
| Subject_6 | 65.24 | 61.12 | 69.29 | 66.60 |
| Subject_7 | 60.56 | 56.83 | 67.02 | 63.08 |
| Subject_8 | 88.65 | 88.54 | 95.66 | 95.64 |
| Subject_9 | 80.48 | 79.65 | 93.42 | 93.14 |
| Subject_10 | 69.92 | 67.44 | 79.33 | 77.27 |
| Subject_11 | 85.00 | 83.78 | 91.13 | 89.84 |
| Subject_12 | 79.44 | 77.92 | 92.10 | 91.98 |
| Subject_13 | 71.19 | 68.63 | 84.64 | 83.62 |
| Subject_14 | 74.05 | 71.89 | 82.97 | 82.18 |
| Summary | 74.24±8.66 | 72.23±9.88 | 82.60±9.56 | 81.16±10.67 |

Table 29: Dataset 2 (5class)

| Subject | Cross Acc (%) | Cross F1 (%) | In-time Acc (%) | In-time F1 (%) |
|---|---|---|---|---|
| Subject_1 | 23.49 | 19.21 | 26.75 | 23.50 |
| Subject_2 | 26.89 | 23.82 | 31.83 | 28.36 |
| Subject_3 | 22.83 | 17.42 | 24.84 | 22.49 |
| Subject_4 | 23.17 | 19.59 | 27.46 | 24.57 |
| Subject_5 | 22.51 | 19.38 | 26.59 | 23.51 |
| Subject_6 | 21.49 | 18.11 | 25.32 | 22.20 |
| Subject_7 | 20.29 | 16.31 | 22.30 | 17.85 |
| Subject_8 | 28.41 | 24.72 | 36.51 | 33.96 |
| Subject_9 | 24.38 | 20.03 | 29.60 | 26.68 |
| Subject_10 | 22.54 | 18.02 | 28.57 | 25.03 |
| Subject_11 | 29.65 | 26.66 | 34.21 | 29.99 |
| Subject_12 | 25.24 | 22.03 | 29.44 | 26.42 |
| Subject_13 | 21.40 | 17.06 | 26.35 | 22.58 |
| Subject_14 | 21.62 | 18.29 | 24.44 | 20.33 |
| Summary | 23.85±2.68 | 20.05±3.00 | 28.16±3.77 | 24.82±3.95 |

## H.2 INDIVIDUAL PERFORMANCE AT THE BEST LEARNING RATE

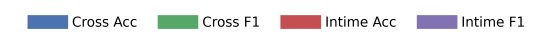

### H.2.1 DATASET 1 (2-CLASS)

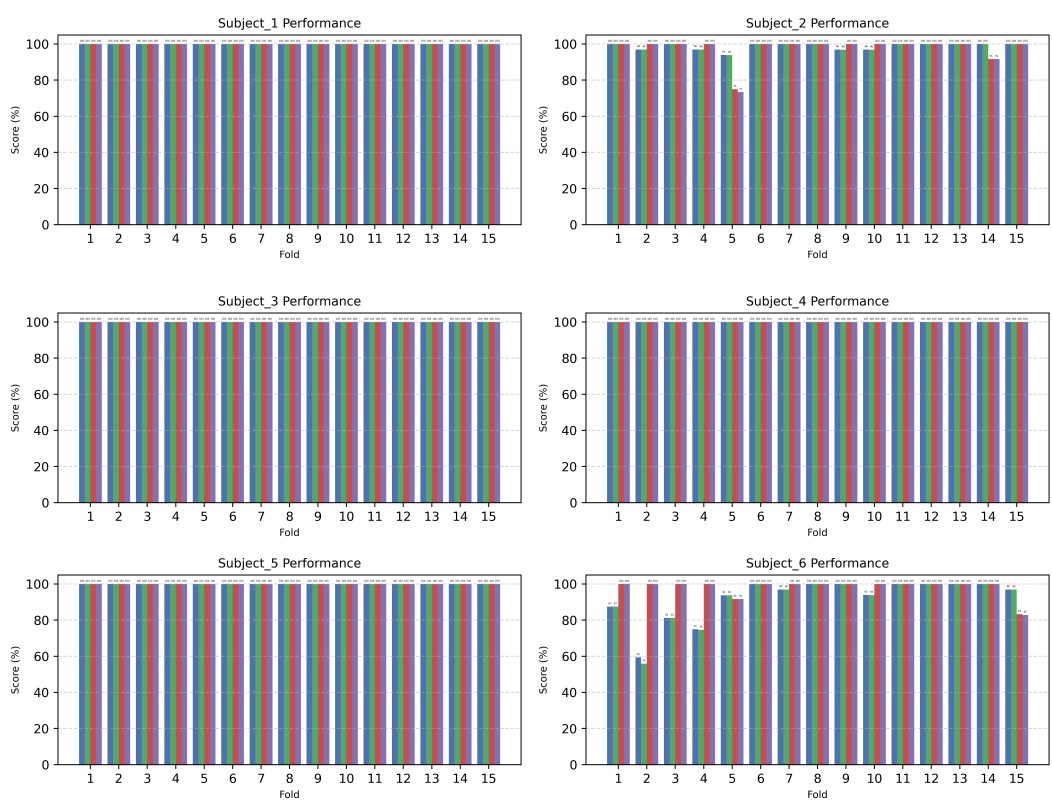

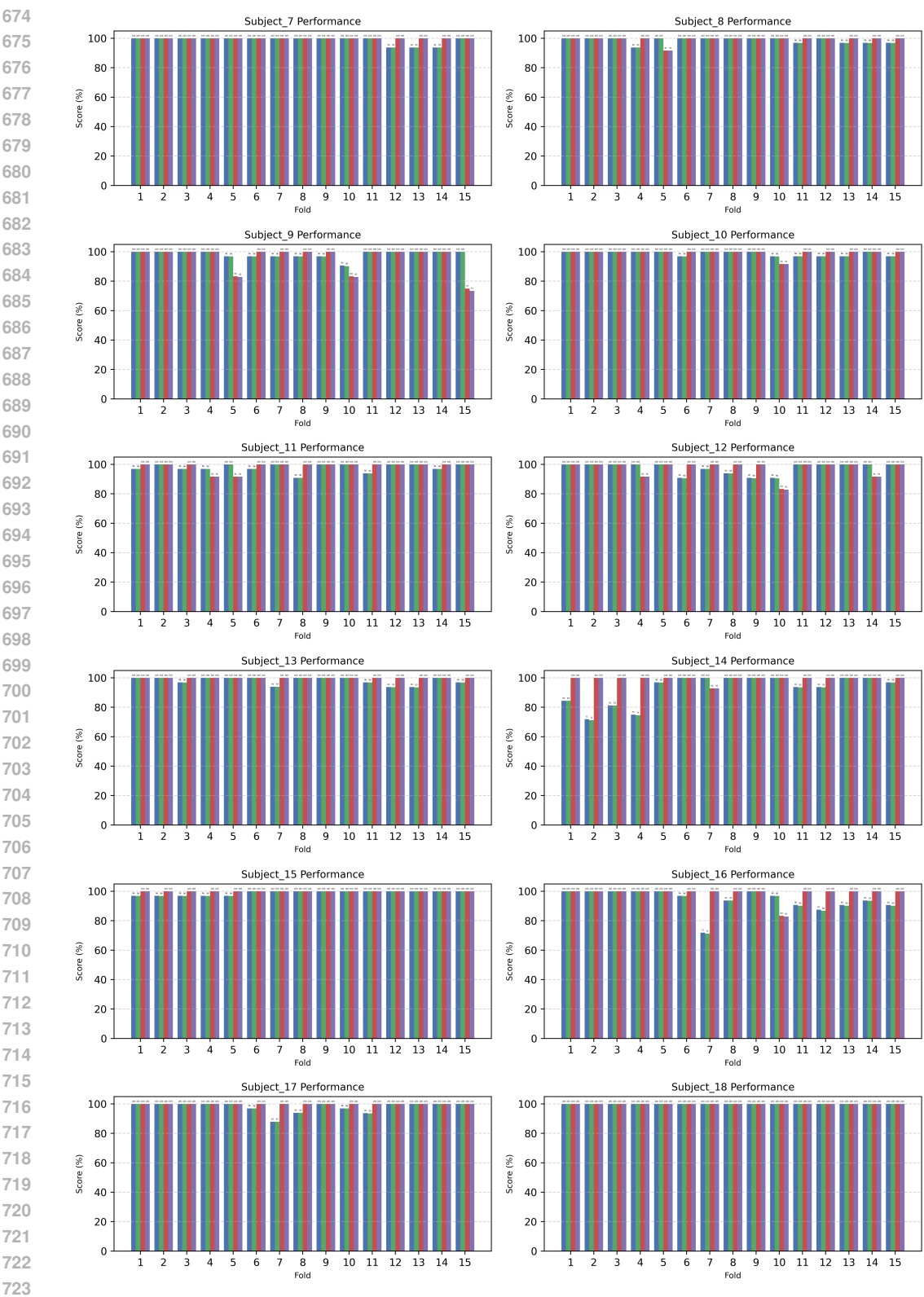

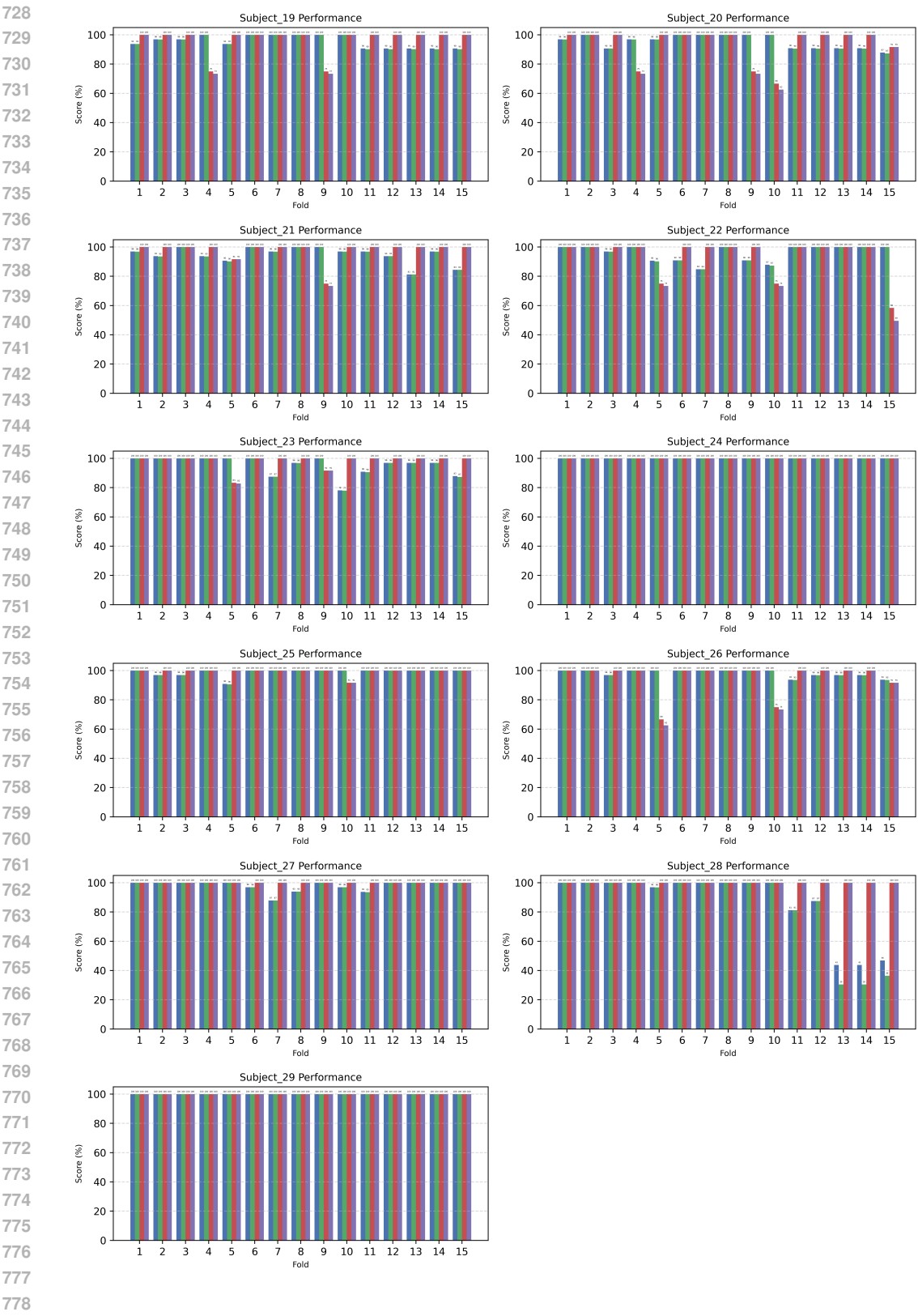

## H.2.2  DATASET 1 (4-CLASS)

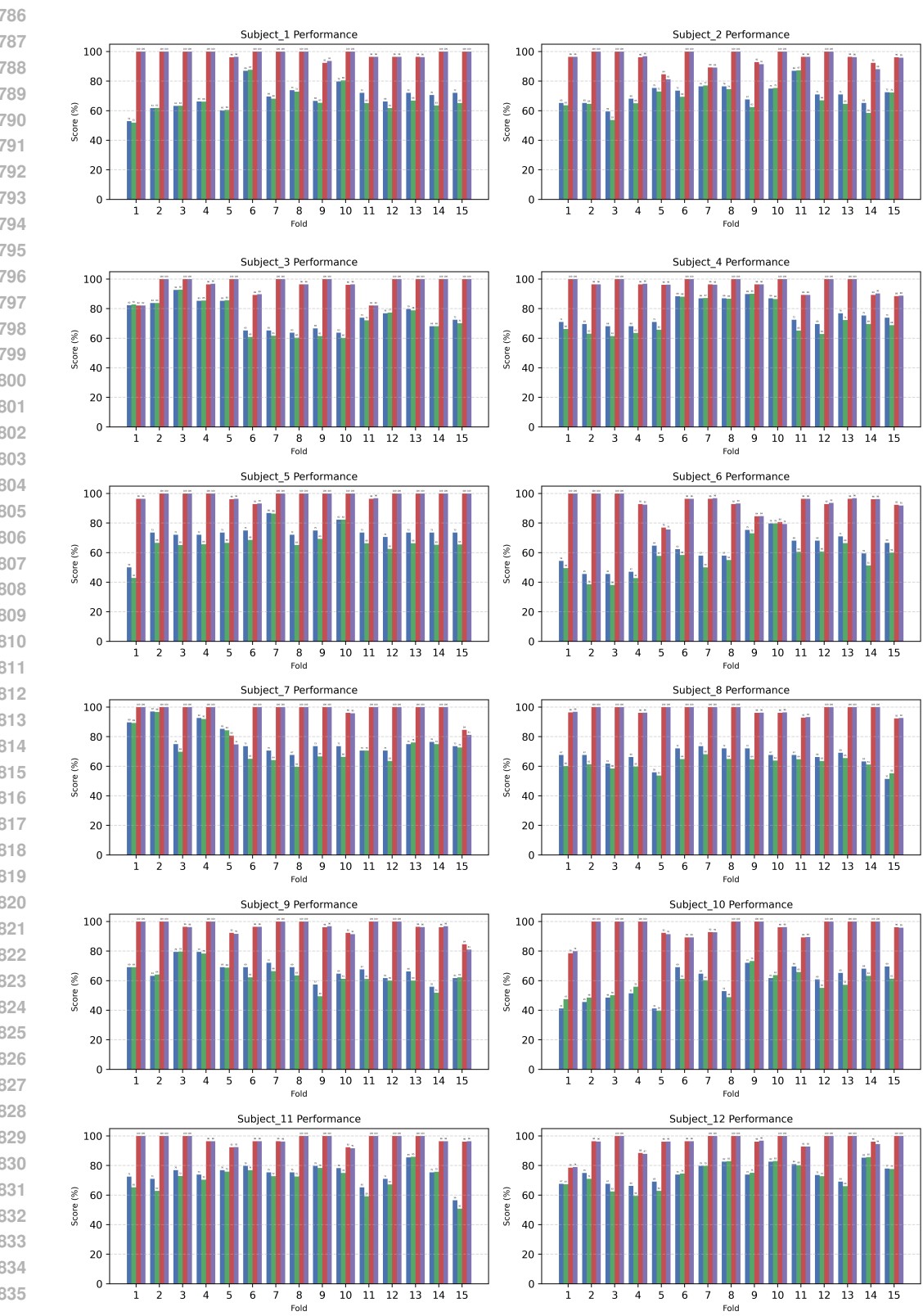

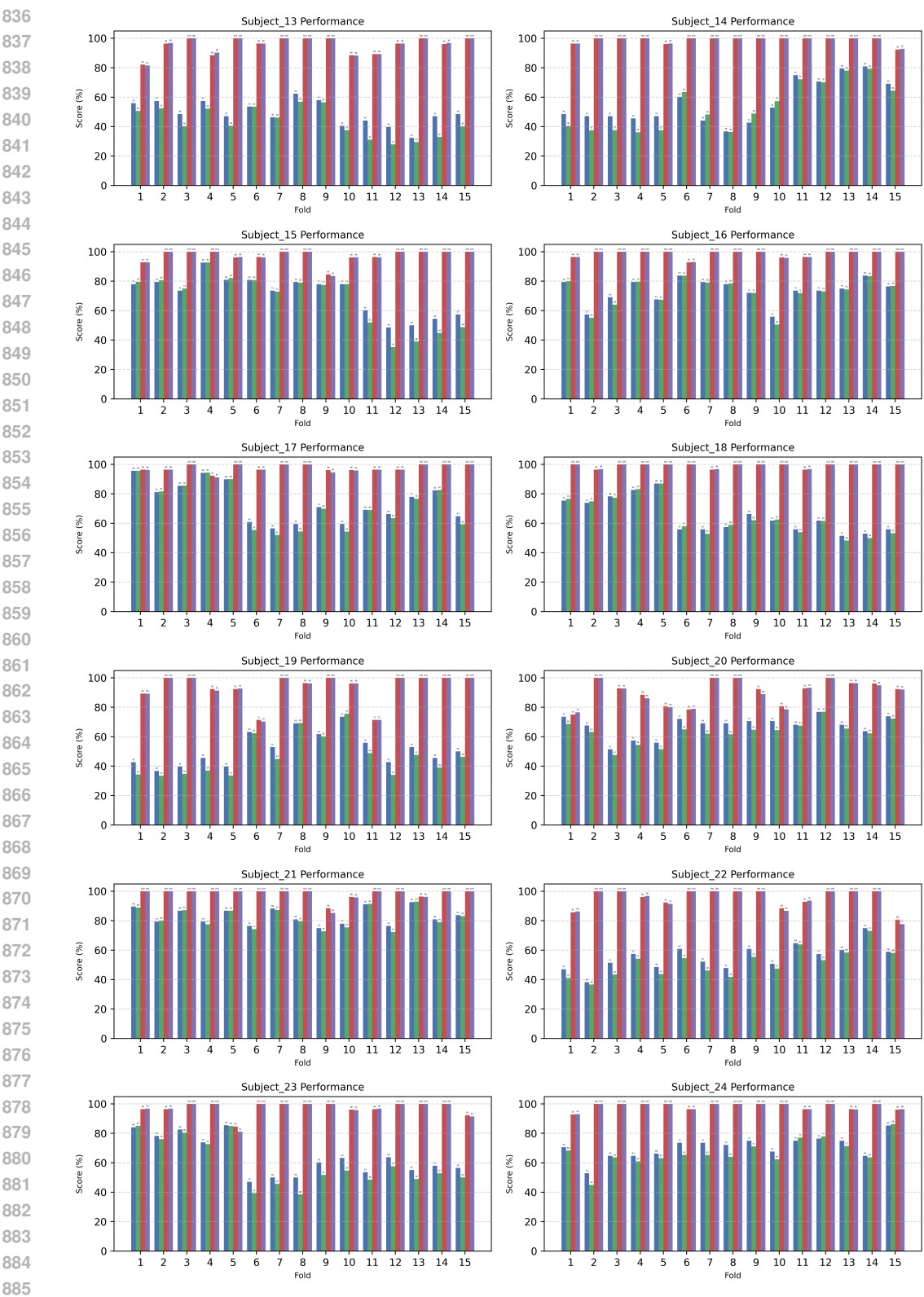

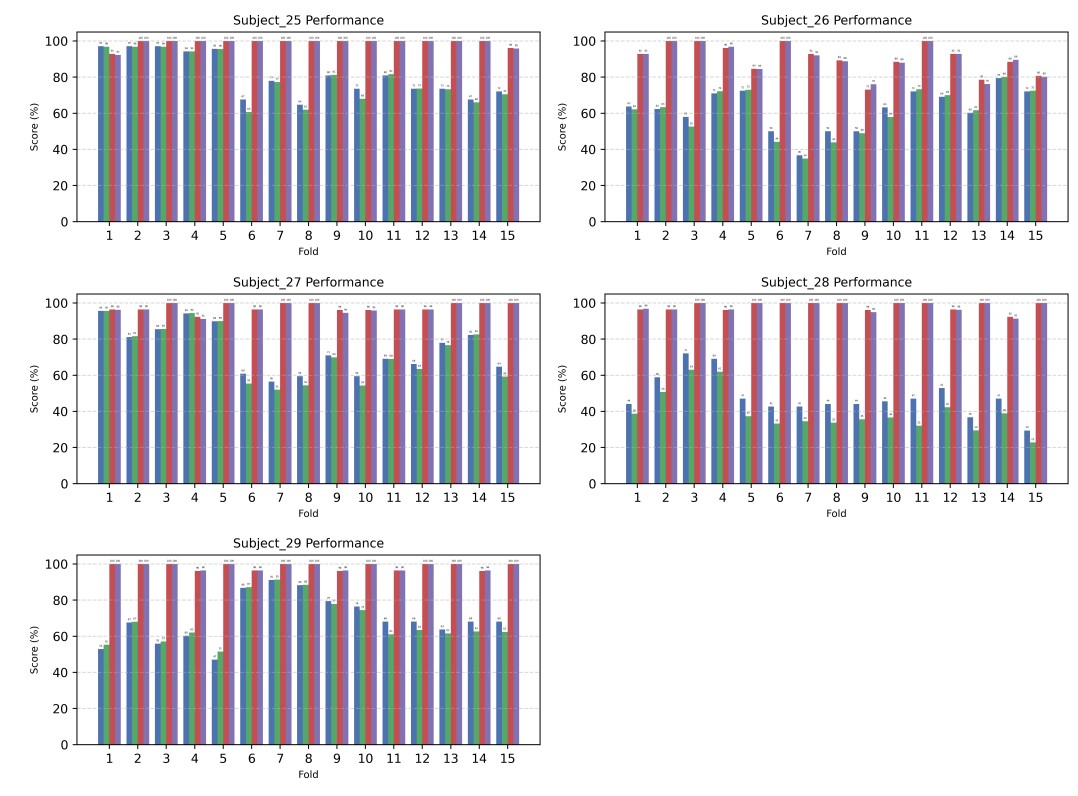

### H.2.3 DATASET 2 (2-CLASS)

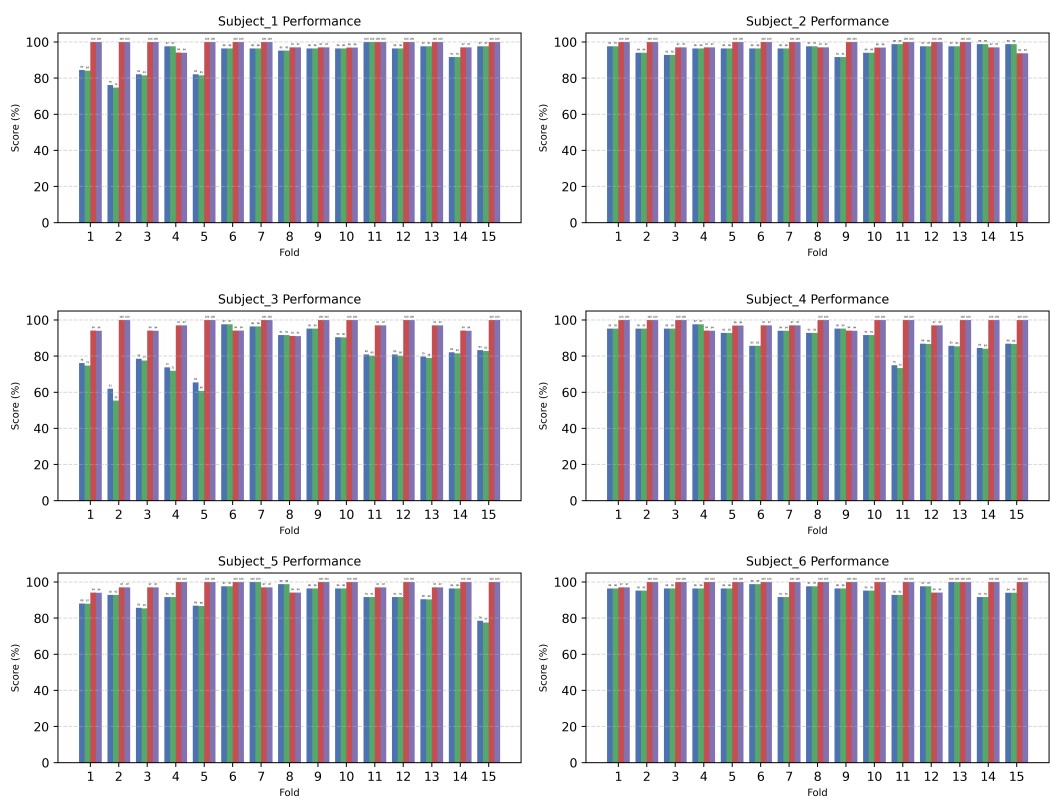

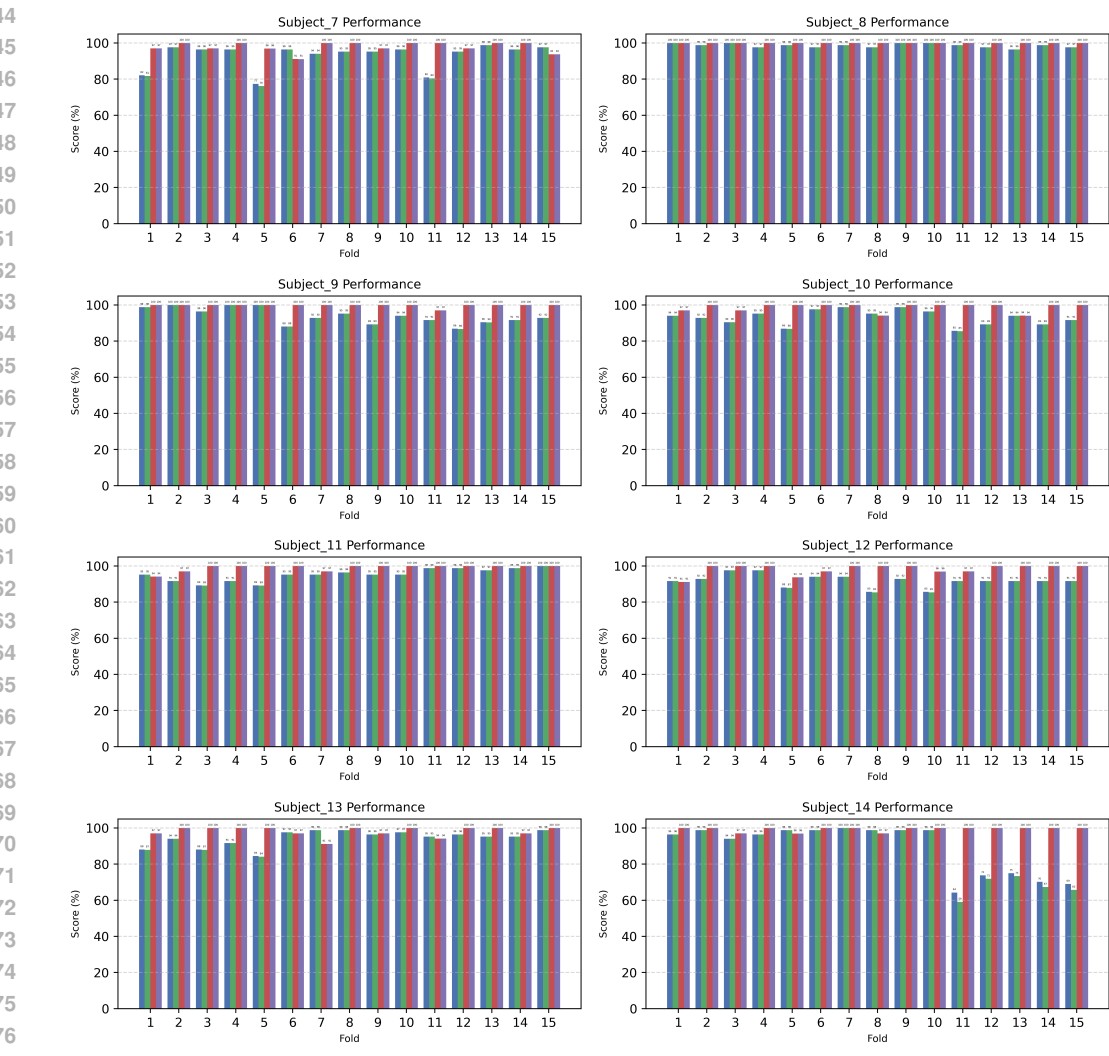

### H.2.4 DATASET 2 (5-CLASS)

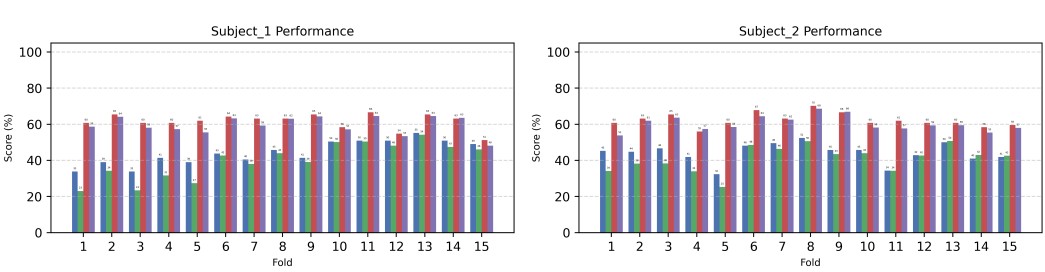

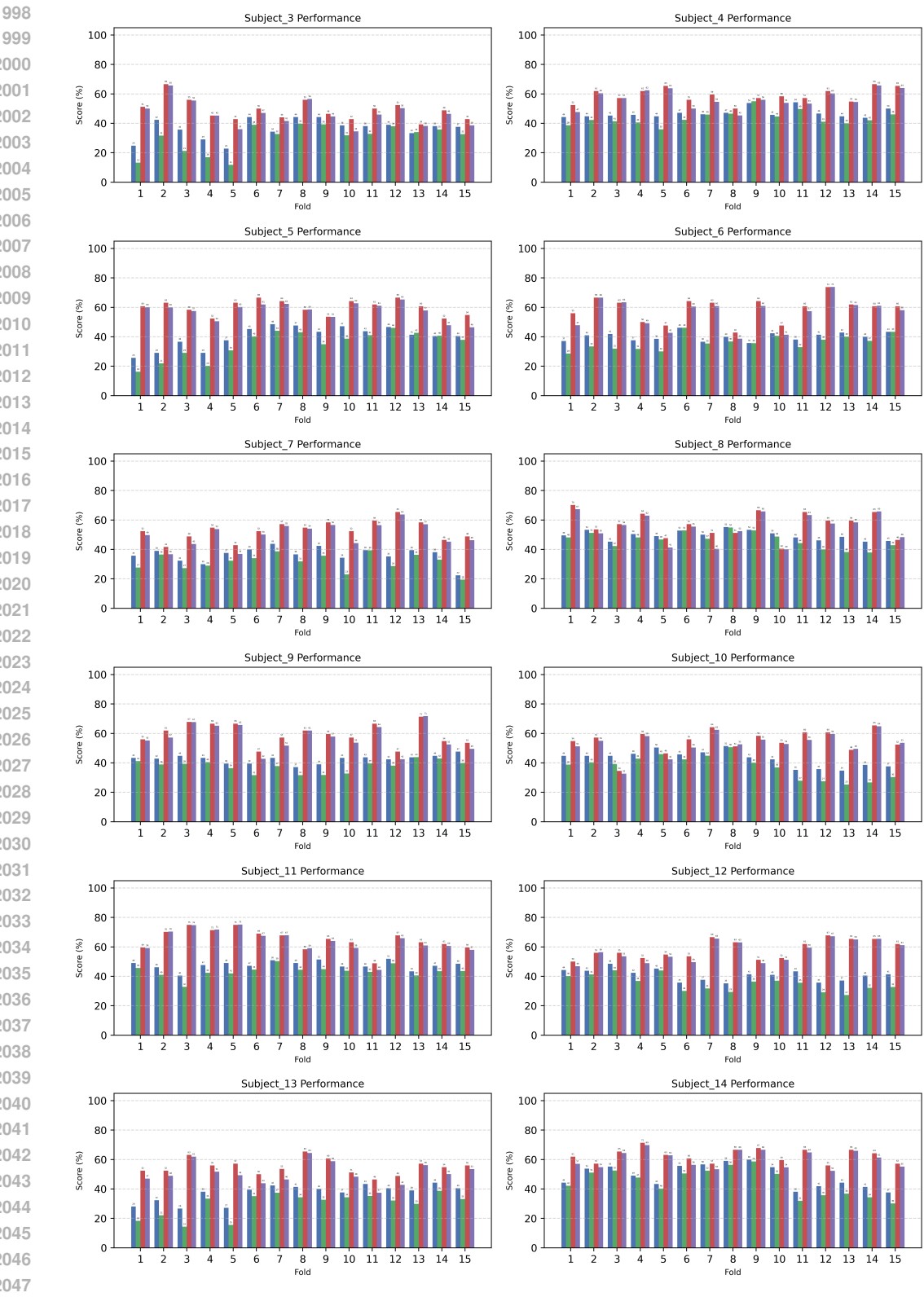

# I   SIGNIFICANCE ANALYSIS

In our study, we fit the following linear mixed-effects model:

$$y_{jk} = \beta_0 + \beta_1 M_{jk} + u_j + f_k + \varepsilon_{jk}, \tag{18}$$

where $y_{jk}$ is the performance metric (Accuracy or F1) for subject $j$ on fold $k$ (of 15), and

$$M_{jk} = \begin{cases} 0, & \text{GACET}, \\ 1, & \text{comparator model}. \end{cases}$$

Here $u_j \sim \mathcal{N}(0, \sigma_u^2)$ is the subject-level random intercept capturing between-subject variability; $f_k \sim \mathcal{N}(0, \sigma_f^2)$ is the fold-level random effect controlling for variability across the 15 folds; $\varepsilon_{jk} \sim \mathcal{N}(0, \sigma^2)$ is the residual error.

We test the null hypothesis

$$H_0 : \beta_1 = 0 \quad \text{versus} \quad H_1 : \beta_1 \neq 0 \tag{19}$$

At a significance level of $\alpha = 0.05$, all analysis results are available in the `log/significance_analysis` directory.

