# OpenReview forum: "GACET: A Graph-Aware Cross-Domain EEG Transformer with the CD-MWL Benchmark"
_ICLR.cc/2026/Conference — ICLR 2026 Conference Withdrawn Submission_

### Official Review · Reviewer_cxJ8 · 2025-10-25

**Soundness:** 2
**Presentation:** 1
**Contribution:** 2
**Rating:** 2
**Confidence:** 3

**Summary:**

The authors introduced an EEG-based mental workload (MWL) assessment dataset comprising 14 participants, which remains relatively small compared to established EEG benchmarks in other domains, such as motor imagery (e.g., the EEG Motor Movement/Imagery Dataset, https://archive.physionet.org/pn4/eegmmidb/, with 109 participants). In addition to the MWL dataset, the authors proposed a novel architecture, a graph-aware cross-domain EEG Transformer, that integrates spectral-temporal features through topology-aware graph reasoning. However, no explainability is illustrated in the current manuscript, though the authors claimed that their model’s learned connectivity patterns align with established neuroscience.

**Strengths:**

[1] A new dataset for MWL (5 difficulty levels) with a detailed data collection procedure\
[2] Cross-day evaluation and cross-subject/leave-one-subject-out evaluation

**Weaknesses:**

[1] Table 1 – The authors are encouraged to specify the number of sessions conducted per participant.\
[2] CD-MWL – The relatively small number of participants in the proposed dataset limits the ability to thoroughly evaluate both subject-independent and subject-dependent cross-session performance.\
[3] Cross-day generalization – The authors should clearly clarify how cross-day generalization differs from cross-session generalization.\
[4] Related Works – The discussion of existing deep learning approaches for brain state decoding is limited. The authors are encouraged to include additional relevant studies to provide a more comprehensive overview of the field.\
[5] Figure 2 – It is recommended that the authors visually demonstrate the dynamic adjacency formulation in Figure 2 rather than only presenting mathematical equations.

**Questions:**

[1] What exactly are Feature 1 and Feature 2 in Figure 2? The authors are encouraged to clearly indicate the type of the feature, e.g., frequency, time, etc.\
[2] What exactly is the built graph? The quality of the manuscript should be substantially improved.\
[3] Many notations/parameters are unclear in the manuscript. For example, how are the “temperature/gating parameters” learned in Equations 7 and 8?\
[4] Why did the authors add Gumbel noise while building the graph? Why didn’t the authors use a fixed graph for a fixed EEG nodes? Equation 10 is very confusing. The authors should provide more details.

**Details Of Ethics Concerns:**

The authors conducted human experiments with high workload conditions. Side effects, such as depression, stress, and unhappiness, should be considered, or, a non-harmful acknowledgement from all participants should be provided.

---

### Official Review · Reviewer_WKV3 · 2025-10-30

**Soundness:** 3
**Presentation:** 3
**Contribution:** 2
**Rating:** 4
**Confidence:** 4

**Summary:**

The paper introduces CD-MWL, the largest public EEG dataset to date for cross-day mental-workload (MWL) assessment, comprising 42 hours of EEG recordings from 14 participants over 3 days. Building on this benchmark, the authors propose GACET (Graph-Aware Cross-domain EEG Transformer), an interpretable neural architecture designed to improve cross-day and cross-subject generalization in brain-computer interface (BCI) tasks.

GACET integrates three core modules: 1. Spectral-temporal feature extraction via differential entropy and sample entropy. 2. Axial Attention Fusion (AAF) to bidirectionally fuse features across frequency and entropy domains. 3. Dynamic Graph Transformer Conv (DGTConv), which learns task-specific functional connectivity using a Gumbel-Softmax-based adjacency construction and topology-aware message passing.

Extensive experiments across multiple EEG datasets (COG_BCI and CD-MWL) and tasks (MATB-II and N-back) demonstrate that GACET outperforms recent SOTA baselines (MAET, TSLANet, LaBraM, SFT-Net, MCA) by 1–4 % in accuracy and F1 under both cross-day and cross-subject settings. Ablation studies confirm the complementary roles of AAF and DGTConv, and connectivity analyses show that the learned graphs align with known fronto-parietal neural networks, supporting the claim of neuroscientific plausibility.

**Strengths:**

1. The CD-MWL dataset expands existing EEG MWL corpora in duration and session diversity, addressing the lack of standardized cross-day evaluation benchmarks.

2. The integration of Axial Attention Fusion and Dynamic Graph TransformerConv blends feature-fusion and learned-connectivity reasoning in a unified, end-to-end framework.

3. Comprehensive baselines, cross-day/subject protocols, and public release of code and logs.

4. The writing is accessible, figures are informative, and the methodological flow is coherent and easy to follow.

**Weaknesses:**

1. The citation in L466 indeed supports frontal-parietal connectivity as a core cognitive-control hub, but does not discuss central-region involvement or occipital connectivity under visual load. The Section 4.5 claim should therefore be narrowed to frontal and parietal regions or augmented with additional references addressing central and occipital roles.

2. The authors claim that their benchmark and model exhibit strong generalization capabilities. However, the evaluation is limited to within-task (MWL) and primarily cross-time generalization. To substantiate the claim, additional experiments demonstrating cross-subject or, ideally, cross-dataset generalization are needed to validate broader transferability across EEG domains.

3. Omission of recent GNN-based BCI models [1] weakens the “graph-aware” claim.


[1] LGGNet: Learning From Local-Global-Graph Representations for Brain–Computer Interface

**Questions:**

1. Are the learned adjacency matrices consistent across subjects and runs?

2. Does “Cross-Domain” mean cross-day, cross-subject, or cross-task?

3. Could you provide a quantitative metric (e.g., ROI correlation) showing alignment with known FPN connectivity?

4. How does DGTConv’s computational cost compare with standard attention layers for real-time BCI use?

---

### Official Review · Reviewer_7NFo · 2025-11-01

**Soundness:** 3
**Presentation:** 2
**Contribution:** 3
**Rating:** 4
**Confidence:** 4

**Summary:**

This paper proposes a model (GACET) and a dataset (CD-MWL) to address the critical challenge of cross-day generalization in EEG-based mental workload estimation. The dataset aims to solve the challenges that existing EEG datasets are limited by short durations, low channel counts, or a lack of cross-day data. The proposed GACET aims to solve the problem that traditional models for EEG decoding fail to capture the brain’s dynamic functional connectivity. GACET integrates spectral-temporal feature extraction (DE + SampEn), axial attention fusion (AAF), and dynamic graph Transformer convolution (DGTConv). Empirical studies on the proposed datasets show that GACET outperforms SOTA baselines (e.g., LaBraM, TSLANet, MAET).

**Strengths:**

- The contribution is multi-facet, including a novel dataset and an interpretable framework.

- The authors released the CD-MWL dataset, covering 14 subjects, 3 days, and 42 hours of EEG mental-workload data. This dataset fills the gap in public benchmarks for multi-day/cross-day EEG mental-workload estimation research. The data, code, training logs, and one-click reproduction scripts are all open-source, enhancing research reproducibility.

- The proposed GACET model not only achieves strong performance in cross-day/cross-session tasks but also provides interpretable brain-network structures via dynamically learned graphs.

**Weaknesses:**

- In section 4.5, the author provides evidence that the importance of brain regions learned by GACET aligns with neurophysiological prior. However, it is different from functional connectivity, which reflects how the information should be gathered from different regions. Hence, it does not mean that GACET is Interpretable, as most existing EEG models tend to assign higher weights to task-relevant brain regions.

- The authors validate the model on their own dataset (Dataset 2) and public dataset (Dataset 1) and in different settings (cross-time, cross-day, cross-subject). The author has addressed their difference in lines 101-108. However, the experimental conclusions and setups are highly similar, making it difficult to justify the necessity of constructing their own dataset.

- The experiments still lack in-depth analysis. For example, how is the performance in different levels of tasks (may lack a confusion matrix) ?

**Questions:**

- Could you further explain why, in Table 5, M4 underperforms M2 and M3? What is the meaning of "module mismatch"?
- According to the performance across different subjects, the average performance of the proposed method depends more on a few low-performance subjects. Does this mean that the proposed method fails with certain subjects, or that all methods perform poorly on these subjects? Would such a situation undermine the significance of designing this method and affect the evaluation results?
- Could you further explain the different settings, e.g., N-back and cross-time? I suggest summarizing different settings in a figure or in a table.

**Details Of Ethics Concerns:**

Privacy and security of the EEG data collected.

---

### Note · Authors · 2025-11-12

I have read and agree with the venue's withdrawal policy on behalf of myself and my co-authors.